# Global Phosphorus Recovery from Wastewater for Agricultural Reuse

Dirk-Jan D. Kok[1*], Saket Pande[1], Jules B. van Lier[1], Angela R.C. Ortigara[2], Hubert Savenije[1], Stefan Uhlenbrook[1,2]

[1] Department of Water Management, Delft University of Technology. Delft, Netherlands.
[2] World Water Assessment Programme, UNESCO. Perugia, Italy.

5 *Correspondence to*: Dirk-Jan Kok (D.D.Kok@cml.leidenuniv.nl)

**Abstract.** Phosphorus is a nutrient necessary for the development of crops and is thus commonly applied as fertilizer to sustain agricultural production. It occurs naturally, at indefinite quantities of uncertain quality in phosphate rock formations, but also accumulates in urban and livestock wastewater wherefrom it is often lost as a pollutant. Recovering phosphorus from wastewater , however, is feasible through struvite crystallization technologies and has the potential to reduce 10 phosphorus pollution of the environment as well as lower the agricultural demand for artificial P-fertilizers. In this study, we developed a model to assess the global potential of P-fertilizer recovery from wastewater and to visualize its trade at sub-national resolution. Results show that humans discharge a maximum of 3.7 [Mt] P into wastewater, thereby potentially satisfying 20% of the global fertilizer demand. Provided 2015 market dynamics, however, the model determines that only 4% of this discharge is technologically and economically recoverable in a market that offers cheap rock phosphate products 15 also. The results of this study demonstrate that in the current economic context, phosphorus recovery from wastewater offers only a small contribution to resolving global phosphorus issues. Nevertheless, this recovery offers many wastewater treatment facilities the opportunity to contribute to creating sustainable communities and protecting the environment locally, while reducing their own operational costs.

**Key words:** *Phosphorus, circular economy, wastewater, struvite, fertiliser*

20 **1 Importance of Phosphorus**

Phosphorus (P) is an element necessary for the development of all living beings as it forms an unsubstitutable, key structural component of DNA and RNA. It is also a limiting nutrient, and therefore growth is often restricted by the lack of P naturally available. While P-related malnutrition in humans is uncommon, inhibited plant growth due to soil phosphorus deficiency is a much more prevalent issue (MacDonald, Bennett, Potter, & Ramankutty, 2011). For this reason, phosphorus is often 25 applied to croplands in the form of organic fertilizers or synthetically as single or triple superphosphate, or mono-Ammonium or di-Ammonium Phosphate (DAP). These fertilizers are easy to transport and distribute over fields, while also readily absorbed by plants. The most essential resource for the production of artificial phosphorus fertilizers is phosphate rock.

30 The rates at which we exploit phosphate rock formations is out of proportion to the rates at which they form - essentially classifying phosphate rock as a non-renewable resource. Peak production of phosphate ore could occur as early as 2030 (Cordell et al., 2009) with economically extractable P resources becoming scarce or exhausted within the next 50 to 100 years (Smil, 2000; Steen, 1998; Van Vuuren et al., 2010). Other authors predict more optimistically that reserves will last

another 300-400 years (van Kauwenbergh, 2010). After this depletion, it will likely be economically infeasible to exploit the remaining ore as it will be too costly to process due to its poor quality (i.e. low phosphorus and high heavy metal content). Gradual depletion of economically extractable reserves will result in further reduction in accessibility to fertilizer by small-holder and subsistence farmers that comprise areas already struggling to cope with food shortages (Pande and Savenije, 2016). Sub-Saharan Africa is one such region, as nearly 75% of its agricultural soils are nutrient deficient, contributing significantly to the crop yield gaps (Sanchez et al., 1997). The prospect of phosphorus depletion ultimately threatens global food security where especially regions of poor soil nutrition levels are vulnerable to its effects.

## 1.1 The Environment, Humanity, and Phosphorus

The introduction of intensified (P) fertilization during the Green Revolution of the 1960's, demonstrated P's significant potential to improve crop yields but also the dangers it poses to the environment. Through seepage and runoff processes (Bouwman et al., 2009), as well as the discharge of improperly treated wastewater (Van Drecht et al., 2009; Morée et al., 2013), phosphorus and other nutrient excesses come into contact with open surface water. As a limiting nutrient, even the smallest quantity of P in water can spark the growth of large algal blooms. These algal blooms have a detrimental effect on aquatic ecosystems, in causing the suffocation of aquatic life through eutrophication, resulting in a loss of habitat and biodiversity (EPA, 2010). Such deoxygenated "dead zones" can be found in both lakes and seas, and affect an estimated 245,000 $km^2$ of marine ecosystems (Corcoran et al., 2010). If excess fertilization and water pollution is a major threat to water quality around the world, then why not extract this excess from the water system and put it back in the food chain? Proper nutrient management practice in tandem with nutrient recovery from rural and urban water systems may potentially be an important strategy to reduce phosphorus pollution by reducing phosphorus discharge to the environment while simultaneously increasing phosphorus supply for food production.

There exist a broad range of sources that contribute the phosphorus loads in wastewater. Excrements, detergents, care products, dishwashing liquids, pharmaceuticals, and food preparation wastes are an example of some of the most significant of these. Aside from this, there also exists a multitude of technologically feasible, phosphorus recovery options (Egle et al., 2016). Unfortunately, these technologies are often deemed too costly to implement and operate while their recovery efficiencies vary, among others, with local wastewater composition and existing wastewater treatment infrastructure. Adoption of these technologies is therefore often challenged by (perceived) economic infeasibility or lacking economic incentives and social stigma. It must be recognized,however, that the economic feasibility of recovery is not globally homogeneous, but varies in space and time. Spatially, the global accretion of phosphorus in wastewater provides recovered products with a (diffused) location-defined competitive advantage over the geographically concentrated rock phosphate mines (fig. 1). While temporally, the appeal for recover will improve over time with the increasing price trends for rock-based fertilizers (e.g. DAP) (fig. 2).[1]

## 1.2 Turning brown waste to green gold

To our knowledge, there exist no studies that evaluate the *spatially dependant* feasibility for phosphorus recovery from

---

[1] Over the past 15 years the phosphorus price of DAP has increased from 665 [$ $t^{-1}$] to 1,552 [$ $t^{-1}$]. In that same period, the price has been as high as 5,217 [$ $t^{-1}$] (2008) and as low as 656 [$ $t^{-1}$] (2002) (IndexMundi, 2017).

wastewater, at sub-national resolution, in a global and economically dynamic context. Insights provided by such a study, however, could accelerate the efficient transitioning to a more sustainable phosphorus fertilizer market by illustrating where recovery may economically be most feasible. We therefore aim to determine the total phosphorus recovery potential from wastewater, as well as the economic feasibility for this recovery, in a global assessment. This is achieved by integrating geospatial data, statistics, and findings from other studies into a model that identifies and connects phosphorus recovery and demand sites based on location, quantities, and prices.

Because of the wide array of pathways to phosphorus recovery, the subject of this investigation is constrained to the recovery of phosphorus from urban and livestock wastewater as a struvite and compost pellet fertilizer product, only. Phosphorus lost via other fluxes (e.g. municipal solid waste) are therefore excluded from this assessment. The sole reason for focusing on recovery through struvite crystallization is because of the fertilizer potential of struvite as well as the current industrial-scale implementation of the technology itself (Cornel and Schaum, 2009).

## 2 Materials and Methods

The phosphorus cycle is delineated by a combination of both social and physical attributes and as such demands a coupled human water systems perspective. A socio-hydrological approach is endeavored as both these attributes are here expressively and emphatically accounted for (Sivapalan et al., 2012). The social component, however, of this coupled human-phosphorus system is confined to the characteristics of a distinct economic nature. The materials and methods employed in assessing the recovery potential of P from wastewater at global scale is therefore interdisciplinary and extensive, covering largely economics within a sciences context.

In general, the methodology can be summarized to consist of three phases which are also used to define the order of the materials and methods section:

1. **Identification of sites and quantities:** Identifying the locations of wastewater accumulation sites and agricultural croplands and assessing the potential associated phosphorus production and demand quantities.
2. **Determination of node prices:** Approximating the minimum production costs of recovering P-fertilizers at the wastewater accumulation (recovery) sites, and the maximum paying prices for P-fertilizers at the agricultural (consumption) sites.
3. **Modelling international trade** in phosphorus, involving:
    a. *Determination of global market price:* Determining an international, free-market price for phosphorus as a function of phosphorus quantities and prices, as well as the distances between the different sites.
    b. *Visualizing trade flows:* Creating a realistic network of P-trade fluxes at subnational resolution.

The main tools employed in this investigation are Geographic Information System (GIS) tools, used to prepare the spatial data, (Q-GIS 2.14) (Quantum GIS Development Team, 2017); and Python 3.6, used to build the trade network model.

## 2.1 Identification of Sites and Quantities

Phosphorus trade occurs between production and demand sites, and therefore two groups of actors are identified in the model: P-producers and P-consumers. The first group, P-producers, consists of three types of actors:

1. **Urban wastewater treatment facilities** – Nodes recovering P from domestic wastewater (i.e. phosphorus excreted, used, and discharged by humans);

2. **Livestock keepers** – Nodes recovering P from animal manure (i.e. manure and liquid waste produced by farm cows, chicken, and swine, while stabled);

3. **Phosphate mines** – Nodes extracting P from rock phosphate reserves.

The three actor types recover phosphorus in different forms, yet all act on the same phosphorus market. The only objective characteristic that the model will use to distinguish one type product from another, is the absolute elemental P value of the product (i.e. U.S. dollars per mass P; [$ mass$^{-1}$] P). Since both mines and recovery sites have the same purpose in the model (i.e. supplying phosphatic fertilizers), both node types are grouped as 'production nodes'.

### 2.1.1 Phosphorus Production

The location and phosphorus production quantities of P-production nodes are determined by integrating geospatial datasets with statistics and findings from other studies. As phosphorus is not 'produced' by organisms, but only consumed and excreted, or used (i.e. detergents), we will refer to the annual amount of phosphorus discharged in wastewater per individual, as the 'phosphorus throughput rate' [kg head$^{-1}$ a$^{-1}$] P. Wastewater phosphorus has variable sources ranging from human excrements to detergents, toothpaste, dishwashing liquids, medicines, food preparation wastes, food leftovers, etc. These sources are not considered individually. Instead we assume that each individual person, globally, contributes an equal contribution to the phosphorus load in wastewater. We combine approximations for these rates from different studies with population density maps for: humans (CIESIN, 2016), cattle, swine, and poultry (Robinson et al., 2014), to determine the spatial distribution of phosphorus excretion rates globally. This spatial distribution of the mass phosphorus produced per unit area, per annum, [kg km$^{-2}$a$^{-1}$], represents the phosphorus production density (a map). Its determination per unit area on this map is summarized for as follows (eq. 1):

$$S = (D * P) * E \qquad\qquad\qquad (1)$$

where $S$ is the maximum organic phosphorus production density [kg km$^{-2}$ a$^{-1}$]; $D$ is the population density [heads km$^{-2}$]; $P$ is the phosphorus throughput rate [kg head$^{-1}$a$^{-1}$]; and $E$ is the estimated base recovery efficiency [-]. For humans, the phosphorus throughput rate ($P$) is assumed globally homogeneous, fixed at 0.77 [kg head$^{-1}$a$^{-1}$]. This is in close relation to other published findings (0.77, Gilmour et al. (2008); 0.78, CRC (2005); 0.2-0.7, Mihelcic et al. (2011); 0.7, Smil (2000)). For livestock, the throughput rate *(P)* is taken to be a function of slaughter weight (FAOSTAT, 2018) following the methodology of Sheldrick et al., (2003). Manure recovery efficiencies *(E)* are taken as: cattle (31%), swine (80%), poultry (77%) (Sheldrick et al., 2003). For humans, it is assumed that 100% of the phosphorus discharged as domestic wastewater reaches the treatment facility. The actual recovery efficiencies will vary per recovery technology implemented at the wastewater treatment plants (WWTP) (section 2.2.). The production density maps for livestock and humans are presented in fig S1a and S1b of the Supplementary Materials. As a final step, the phosphorus production densities per land area are

converted into point nodes using GIS tools (see section 2.1.3). The spatial variation in recovered P-production potential is therefore marked by global variation in population and livestock densities.

The locations for the P-mining industry are, instead, geographically concentrated. Data on and P-production values and mine site locations is acquired from a USGS (2002) dataset. This dataset is adjusted to match the USGS reported phosphate production estimates for different simulation years.

### 2.1.2 Phosphorus Consumption

Similar to the phosphorus production density, the phosphorus demand density map represents per unit area the yearly amount of phosphorus required by agriculture [kg km$^{-2}$ a$^{-1}$]. It is determined following a comparable methodology to that of phosphorus production, where crop densities, as approximated through crop harvested area maps (Monfreda et al., 2008), are related to phosphorus-requirement rates (UNIDO and IFDC, 1998). The crop phosphorus requirement rates [kg km$^{-2}$ harvest$^{-1}$] as reported by UNIDO and IFDC (1998), however, are determined for optimal yield. Provided that no farmer is going to fertilize for optimal yield when he or she knows that these yields are unachievable provided persistent regional water limitations, the actual P demand is proportionally reduced to the potential *water-constrained* yield. This assessment is made for six major crops: maize, wheat, rice, sorghum, soy bean, and potato. The water-constrained yield is determined by adapting the evaporation-transpiration deficit equation (Steduto et al., 2012) to the following (eq. 2):

$$1 - \frac{Y_a}{Y_m} = K_y \left( 1 - \frac{E_a \cdot \frac{T_g}{365} \cdot A_H \cdot C}{E_m} \right) \tag{2}$$

where $Y_a$ is the actual yield [kg km$^{-2}$ a$^{-1}$]; $Y_m$ is the optimal yield [kg km$^{-2}$ a$^{-1}$]; $K_y$ is the crop coefficient [-]; $E_a$ is the cumulative actual evaporation-transpiration per year [mm a$^{-1}$ Area$^{-1}$]; $T_g$ is the duration of the crop growing period [d]; $A_H$ is the fractional area harvested [-] ; $E_m$ is the evaporation-transpiration for optimal yield per harvest [mm a$^{-1}$ km$^2$]; and $C$ [-] is a correction factor. In this investigation, $E_m$ is assumed equal to the crop water requirement for optimal yield. Global approximations of $K_y$ and $E_m$ values were retrieved for different crops from FAO sources (2015), and $E_a$ was approximated from MODIS evapotranspiration products (NASA, 2005). A summary of the data used and its sources is presented in Table S2 of the Supplementary Materials.

Eq. 2 is an adaptation of the original evaporation-transpiration deficit equation. The original equation is designed for a single growing period, yet much of the other input data is yearly. As the beginning and ending of a crop's growing season will vary globally, it is not possible to combine global *yearly* harvest area maps with *monthly* evaporation data. Reformulation was therefore necessary to account for a difference in temporal scales of the input data. For the adaptation, it is assumed that of the yearly evaporation, an amount proportional to the duration of a crops growing period is evaporated during the crop growing season ($E_a \cdot \frac{T_g}{365}$). This value is then further reduced by multiplication with the fractional crop harvested area ($A_H$) to account for evaporation from other landcover types in the area as well. Finally, as these simple manipulations introduce a significant error, a correction factor ($C$) was added to globally scale potential yields greater than optimum (i.e. >1), back down to optimum (=1) and to achieve a total, global phosphorus demand that approaches observed values for these crops.

In this investigation, we assume no change in soil stored phosphorus. The yearly phosphorus demand per area therefore reflects only the crops yearly phosphorus uptake as a function of a crops water-constrained yield and harvested area. This phosphorus demand is described as a linear regression between yield and P-fertilizer requirement through eq. 3:

$$D_{PT} = \frac{Y_a}{Y_m} \cdot A_H \cdot P_{opt}^n \qquad (3)$$

where $D_{PT}$ is the calculated phosphorus demand density [kg ha$^{-1}$a$^{-1}$]; $P_{opt}^n$ is the crop specific (*n*) P-requirement for optimal yield [kg ha$^{-1}$]; and $A_H$ is the crop harvested area [ha$^{-1}$]. The parameters per crop are again summarized in Table S2 of the Supplementary Materials. The six crops evaluated for make up roughly 56% of the global demand (Heffer, 2009). Although spatially inaccurate of the actual global phosphorus demand distribution, the total, global phosphorus demand quantity is approached by dividing each area's value for the six major crops by 0.56 (56%). The demand density maps for agriculture are presented in fig S1c of the Supplementary Materials.

## 2.1.3 From raster to nodes

The areas of major production and consumption densities, determined in 2.1.1 and 2.1.2, are aggregated into nodes that are described by a coordinate position, a class (group: urban or livestock, or crop type), and a quantity of yearly phosphorus supply or demand. Each node is systematically positioned in the centre of a larger area of uninterrupted high phosphorus production or demand density as determined by the raster map calculations preformed in steps 2.11 and 2.12. To avoid the aggregation of administratively separate regions into a single node, the areas of continuous high demand/production density are separated by national boundaries for smaller countries, and first level administrative borders (e.g. states and provinces) for large countries (e.g. U.S.A., India, Russia, Canada, etc.). Nodes with a production values of less than three kilo tonnes per year are considered insignificant in the global context and are therefore excluded from further consideration in the economic analysis. This constrains the total number of actors, reduces the complexity of the network, decreases the processing time, and improves visualization of the results. This preselection reduced the global P-recovery quantity by 15%, while reducing the number of actors by 76%. The trade model thus only accounts for trade from 24% of all potential recovery sites, which, nevertheless, represent 85% of the global recovery potential.

### 2.2 Determination of Node Prices

Throughout this investigation, prices are presented as the price per tonne phosphorus in each fertilizer product. [$ t$^{-1}$]. This is because struvite fertilizers (14% P), conventional artificial fertilizers (DAP: 22% P), and compost pellets (1% P) are, although different in form and P contents, acting on the same phosphorus market. In the model market, they are discriminated only based on their total phosphorus content and, as such, their values are determined based solely on the mass of P that they contain.

Whether trade is possible between a demand and a production node depends on the transportation and production costs of the production node and the maximum bid price of the demand node. Although having identified three different producers (section 2.1), each creating their own unique fertilizer products, they are all subjected to the same economic constrictions.

The production cost of each producer is defined by: i) an investment cost for infrastructure, and ii) a variable cost per mass P recovered. Additionally, they are imposed with iii) a transportation cost for selling. The sum of these costs determines how attractive a producer is to the individual, agricultural demand nodes.

### 2.2.1 Cost for Recovery

The cost for recovery (i.e. production cost) varies depending on the recovery technology whose feasibility for implementation, in turn, depends on wastewater composition and existing infrastructure. In this study, municipal wastewater composition is assumed roughly, globally homogenous. The Sustainable Development Goals (SDG) dataset on *percent of urban population with access to sanitation* is then used to approximate how well developed the existing sanitary infrastructure is in different parts of the world (WHO/UNICEF JMP, 2015). This estimation is then used as an indicator for the feasibility to implement certain recovery technologies. How the SDG dataset is used to interpret the feasibility for implementation of specific recovery technologies, is presented below:

**A high national percentage of urban population with access to sanitation (>90%)** is likely to be indicative of highly developed countries that observe stricter effluent standards and that therefore have, or are working towards upgrading, conventional wastewater treatment plants (WWTP) for biological nutrient recovery (BNR) or chemical phosphorus removal. The phosphorus recovery costs for these highly developed nodes are those associated with the investment in a large Ostara® Pearl Reactor for struvite precipitation. The production cost for this highly developed group is reduced by the savings in uncontrolled struvite scaling maintenance and sludge handling costs that are associated with controlled struvite precipitation (Shu et al., 2006). Although other struvite recovery technologies are available, Ostara® Pearl Reactors were chosen given their commercially effective implementation in various countries. There exist other recovery technologies that allow for absolute greater recovery amounts (see Egle et al., (2016)), but few of these are potentially economically competitive producers of pure P-fertilizer products.

Most of the influent phosphorus at a WWTP accumulates at the centrifuge (80-90%), where it is separated in centrifuge cake (~85%) and liquor (~15%) (Jaffer et al., 2002). Struvite crystallization from centrifuge liquor achieve efficiencies higher than 90% (Jaffer et al., 2002; Münch and Barr, 2001). Therefore, assuming some variability amongst WWTP, we make the optimistic estimate that approximately 20% of influent wastewater phosphorus may be recovered at BNR WWTP through struvite precipitation.

**Areas with intermediate urban access to sanitation (40-90%)** are assumed to be serviced by simple, centralised wastewater treatment facilities. The technology investment cost for these nodes are the same as for the highly developed infrastructure group but excluding this time the sludge handling cost savings. The recovery efficiency is again assumed to equal 20% of the influent P.

**Low urban access to sanitation (<40%)** is taken to be indicative of low sanitary development and thus offers the flexibility to adopt more novel, less water dependant forms of sanitation. The technology applied for these areas are source separating-, dry composting toilets, where urine and faecal compost are collected separately. The urine is collected by 40,000 litre tank trucks and processed at a centralized struvite precipitation facility. Faecal compost is collected, dried and processed into compost pellets also at a central facility. A 90% efficiency is easily achieved when struvite is precipitated from source-separated urine (Wilsenach et al., 2007). It is furthermore assumed that all of the faecal phosphorus is retained in the compostation, drying and pelletization processes.

**For livestock nodes** the collected manure is composted, dried and pelletized, also. As opposed to the source-separated faeces of dry composting toilets, the pelletization of manure from livestock farms occurs not at a centralized facility, but on-site. This is assumed feasible providedthe high volumes of manure produced by livestock in comparison to humans.

For all production nodes, a generalized phosphorus recovery cost can be described as follows (eq. 4):

$$f_{min}^i = \frac{R \cdot S_{PT} + B}{S_{PT}} - S + \frac{T^i}{\rho} \tag{4}$$

where $f_{min}^i$ is the minimum price for phosphorus produced at node $i$ [\$ t$^{-1}$]; $R$ is the variable cost per tonne P recovered (i.e. the magnesium cost for struvite precipitation; the pelletization cost for compost pelletization) [\$ t$^{-1}$] P; $B$ fixed operational cost [\$]; $S_{PT}$ is the total phosphorus recovery potential [t]. Furthermore, where relevant, S is the struvite scaling and sludge handling cost savings per tonne P recovered [\$ t$^{-1}$] P (i.e. for BNR plants); $T^i$ is the intracity transport cost of collection [\$ t$^{-1}$] (i.e. for urine and faeces from source-separating toilets); and $\rho$ is the proportion of phosphorus by weight of the transported material [kg kg$^{-1}$], which are 0.066% and 0.46% for urine and dried, faecal, toilet compost respectively (Vinnerås, 2001). The annual fixed operational costs are taken as the annual costs minus the resource costs as reported in the dissertation of Egle (n.d.), TU Wien. For BNR plants, a struvite and sludge handling cost savings is included of 0.89 [\$ kg$^{-1}$] struvite removed (Shu et al., 2006). This essentially allows BNR plants to supply struvite for free (excl. transportation costs), which common occurrence for struvite precipitating BNR WWTPs in The Netherlands. The additional variable cost for the dry-toilet solution is attributed to the collection of waste and a pelletization cost of \$30 [\$ t$^{-1}$] compost (Masayuki Hara, 2001). A summary of these and other data values are presented in Tables S3, S4 and S5 of Supplementary Materials.

The intracity transport cost $T^i$ is described as followed (eq. 5):

$$T^i = D_{ta} \frac{P_d \cdot E_{Lt} + \frac{L_{ct}}{\bar{V}_{Lt}}}{W_{Lt}} \tag{5}$$

Where $D_{ta}$ is the average distance of a return journey for a tanker truck in servicing the city per full load [km]; $P_d$ is the price of diesel [\$ L$^{-1}$]; $E_{Lt}$ is the fuel efficiency of the tanker truck [L km$^{-1}$]; $L_{ct}$ is the labour wage [\$ head$^{-1}$]; $\bar{V}_{Lt}$ is the average velocity of the truck in the city [km head$^{-1}$]; $W_{Lt}$ is the truck load weight [t]. The transportation cost from the processing facility to the consumer is included in the final sale price [\$ t$^{-1}$] of the producer and is therefore not accounted for yet at this stage.

**2.2.2 Maximum Buyer Bid Price**

The maximum price that demand nodes are willing to purchase phosphorus at depends on the marginal value of phosphorus. This varies per crop type and can be described as followed (eq. 6):

$$f_{max}^n = \frac{Y_{opt}^n \cdot C_a^n}{P_{opt}^n} \cdot R^n \tag{6}$$

where $f_{max}^n$ is the maximum price for phosphorus [$ t$^{-1}$]; $Y_{opt}^n$ is the optimal yield [t ha$^{-1}$]; $C_a^n$ is the crop price in year $a$ [$ t$^{-1}$]; $P_{opt}^n$ is the optimum fertilizer dosage rate (equal to total P-requirement for optimal, water constrained yield) [t ha$^{-1}$]; and $R^n$ is the ratio of fertilizer cost to total production costs [-], for crop $n$.

### 2.2.3 Transportation cost

Lastly, the transportation costs between the production and demand sites are determined with *as-the-crow-flies* distances and the parameters given in Table S3 (Supplementary Materials) substituted into the following transport cost equation (eq. 7):

$$T_c^{i,n} = D^{i,n} \left[ F_W \cdot \frac{P_b \cdot E_W + C_F}{\bar{V}_W \cdot W_W} + F_L \cdot \frac{P_d \cdot E_L + \frac{L_C}{V_L}}{W_L} \right] \tag{7}$$

where $T_c^{i,n}$ is the transportation cost from node i to node n [$ t$^{-1}$]; $D^{i,n}$ is the distance between node $i$ and node $n$ [km]; $P_b$ is a bunker fuel price (shipping fuel) [$ t$^{-1}$]; $P_d$ is the price of diesel [$ L$^{-1}$]; $E_W$ is the fuel efficiency of a container mainliner ship [t d$^{-1}$]; $E_L$ is the fuel efficiency of a 2x30 tonne truck combination [L km$^{-1}$]; $C_F$ is the fixed costs per ship [$ d$^{-1}$]; $L_c$ is the labour wage [$ head$^{-1}$]; $\bar{V}_W$ is the average velocity of the ship over water [km d$^{-1}$]; $\bar{V}_L$ is the average velocity of the truck over land [km head$^{-1}$]; $W_W$ is the full ship weight [t]; $W_L$ is the full truck weight [t]. $F_W$ and $F_L$ are the fractions of the total distance that is travelled over land and over sea. The model at present does not distinguish between the transportation over land and over sea based on observed geography. Instead the model employs a cumulative probability curve that approximates the proportion of the total distance likely to have been traversed over water, $F_W$ (eq. 8); and over land, $F_L$ (eq. 9). It is assumed that at least 15% of the total distance is always transversed over land.

$$F_w = \frac{85}{1 + e^{-\frac{D - \mu}{S}}} \tag{8}$$

$$F_L = \frac{85}{1 + e^{\frac{D - \mu}{S}}} + 15 \tag{9}$$

where μ and S are function shape constants of 500 and 100 [-], respectively.

### 2.3 Trade model

A model is constructed to determine i) the market price for international phosphorus trade, and ii) what amounts of phosphorus are being traded between which nodes. Market prices emerge as a function of the individual production/consumption prices and associated supply/demand quantities, which will vary depending on which actors are included in the market scenario. The phosphorus recovery potential is therefore assessed for three different combinations of actors that represent, respectively, the current, potential near-future, and potential far-future markets:

1) **Current market - mine supplied products only:** The current phosphorus market is strongly rock-phosphate oriented. When the model runs the data for a 'current market' scenario, only rock phosphate products are available on the market. This scenario serves mostly a model validation purpose.

2) **Future market - both mine and recovered products:** Recovered phosphorus may become a more important product in the future market. When the model runs the data for a 'future market' scenario, it is assumed that both rock phosphate as well as recovered phosphorus products partake.

3) **Far-Future market - only recovered products:** In the far future market, most rock phosphate reserves will have been depleted. When the model runs the data for a 'far-future' scenario, it is assumed that rock phosphates no longer take part in the market which is then solely dominated by sustainable, recovered products.

4)

### 2.3.1. Determination of global market price

The global market price is determined as the price at which total quantity of phosphorus demanded (sum of agricultural demand quantities) is equal to the quantity supplied (sum of P-production quantities). It is approximated as the point where global demand function (defined as cumulative phosphorus demand vs. maximum buying price), intersects the global supply function (cumulative phosphorus production vs. production price). The demand function is the locus of the maximum prices at which demand nodes are willing and able to purchase phosphorus, and the supply function is the locus of minimum prices at which supply nodes can sell certain amounts of phosphorus without going out of business (i.e. without making a loss). Where the two curves intersect, the market for P is cleared providing a best approximation of the market price (Arrow and Debreu, 1954). The Supplementary Text provides an illustration of how market prices for the three scenarios are determined following this principle, and how transportation costs complicate this method of price determination.

### 2.3.2. Quantification of Trade Flows

The quantity of phosphorus traded for each scenario is determined following a method of reduction and elimination. Firstly, a list of all possible combinations of supply and demand nodes is created. Each combination of supply and demand nodes is passed through two 'filters' that removes some pairs provided simple conditional statements. This reduces the list down to a selection of trading node pairs for each market scenario, for each year.

Before anything, the first filter in the model removes the pairs that can never trade with each other based on their combination of the minimum production costs, the maximum bidding price, and transportation costs associated with the data of that year (e.g. fuel cost, wastewater flows, etc). The second filter then removes node pairs which cannot trade with each other at a given 'hypothetical market' price imposed on the network. Either the production cost may be above-, or the maximum bidding price may be below the imposed market price, implying that the nodes cannot trade. If both the production cost is below- and bidding price is above the imposed market price, then the node pair is left in the list for that 'theoretical' market price. After these two filters, the list of potential trade partners is reduced significantly and assessment may be made as to whether the saved pairs actually trade and then with what quantities.

In the model, phosphorus consumers will look for the cheapest suppliers. The matter becomes obscure here as, in reality, there are no cheaper or more expensive suppliers for a single market price. Supply nodes that could, however, supply at prices far lower than the set market price (due to lower production costs) have a competitive advantage over those that cannot. At the same time, agricultural demand nodes that are willing to pay much more than the hypothetical market price, have a greater financial capacity to outbid those agricultural nodes whose maximum bid price is much closer to the

hypothetical market price. The difference between this market price and i) the production and transportation costs for production nodes and ii) maximum bid price for demand nodes, shows how competitive a node pair is. If a node is *able* to produce and transport at prices much lower than market price, and if a demand node is *able* to pay much more than the market price, then the model assumes that trade between these most competitive nodes occurs first. Therefore, the list of

5  remaining node pairs is sorted according to the greatest difference between production + transport cost, and maximum bid price, with market price.

For each of node pair lists for the different 'hypothetical market' prices, trade is executed and the list updated. After each trade, the list is updated in terms of total phosphorus quantity ($Q$) demanded by demand node ($D$) number *n,* or supplied by

10  supply node ($S$) number $i$ ($D_Q^n$, and $S_Q^i$, for demand and supply respectively). The amount traded ($Q_{(i,n)}$) between each node pair is taken to be equal to the minimum of supply or demand as formulated below (eq. 10):

$$Q_{(i,n)} = \begin{cases} S_Q^i & if\ D_Q^n > S_Q^i; \\ D_Q^n & if\ S_Q^i > D_Q^n; \end{cases} \tag{10}$$

The supply available and quantity demanded at each supply demand nodes are updated as follows (eq. 11),

$$
\begin{aligned}
S_Q^i &= 0, & D_{Q(new)}^n &= D_{Q(original)}^n - Q_{(i,n)} & if\ D_Q^n > S_Q^i; \\
D_Q^n &= 0, & S_{Q(new)}^i &= S_{Q(original)}^i - Q_{(i,n)} & if\ S_Q^i > D_Q^n;
\end{aligned}
$$
$$\tag{11}$$

By eq. 11, one of the nodes will have 0 production capacity or demand after each trade, and so all possible trade combinations with that node are removed from the list of possible trade partners remaining for that hypothetical market price. The trade is recorded in a separate list of 'successfully executed trades'. This process is continued until the list is empty and thus all feasible trade for that price has been conducted. Plotting the cumulative phosphorus trade for each hypothetical market price simulated, results in a combined version of the supply and demand curves of 2.3.1, where the apex

coincides with the determined market price. The trade pairs saved in the list of 'successfully executed trades' are then connected through a series of coloured vertices on maps to visualize the trade network.

### 2.4. Simulation years

The model is setup using human population density, livestock population density, and croplands data for 2005. This data is adjusted using constant growth rates when simulating other years. For example, in extrapolating the human population

density (CIESIN, 2016), and therefore phosphorus production potential, from 2005 to another year, the models employs the following equation:

$$P_y = P_{2005}(1 + 0.0122)^{y-2005} \tag{12}$$

Where $P_y$ is the population density for an area in year *y;* $P_{2005}$ is the population density for an area in 2005; and 0.0122 represents the 2005 to 2015 averaged, global human population growth rate of 1.22% per year (World Bank, n.d.). The social

phenomena of migration and urbanization are not considered in the process. Also, contrasting growth rates of the different areas of the world are lost, as by this definition the population in each node, everywhere, grows equally. A summary of this yearly data and its sources is presented in Table S4 of the Supplementary Materials. This is followed up by Table S5 which presents an overview of most fixed parameters.

## 3. Results

### 3.1. Phosphorus recovery potentials excluding economic dynamics

The phosphorus recovery potential can be assessed in three different ways depending on the constraints imposed. Firstly, and most simply, the sum of all global production densities (2.1.1) provide an indication of the *total potential* of recovering all
10 excreted phosphorus, everywhere, without regard for any economic dynamics. For 2015, these are determined to amount to 3.7 [Mt a$^{-1}$] P for humans, and 17.39 [Mt a$^{-1}$] P for livestock. Recovering all urban phosphorus discharged as wastewater can thus potentially satisfy 20% of the 19.1 [Mt a$^{-1}$] calculated agricultural demand. Recovering all phosphorus in animal manure can potentially satisfy 90% of the total agricultural demand. The recycling of all animal manure and human excreta confirms the large potential of recovery to substitute phosphorus fertilizers (Bouwman et al., 2009).

Unfortunately, however, it is not feasible to recover every ounce of phosphorus excreted or to fertilize every crop patch everywhere. More realistically, recovery will be efficient in areas of high population or livestock density, while fertilisation will benefit mainly areas of intensive agriculture. By imposing a size constraint on the nodes determined from the phosphorus density maps (section 2.1.3), a more realistic assessment of the contribution of recovered products to the global
P demand can be made. For 2015, the phosphorus recovery potential from high density urban sites is estimated at 1.73 [Mt a$^{-1}$] P, which is 48% of the original 3.7 [Mt a$^{-1}$]. This percentage approaches the percentage of global population urbanized, 54% (World Bank, 2018), partially confirming that this constraint indeed excludes smaller rural areas, thereby limiting recovery to the high potential, urban sites only. Recovery from livestock is reduced from 17.39 to 8.8 [Mt a$^{-1}$] P, now accounting only for the most intensive animal husbandry sites in the world. With a demand of 16.81 [Mt a$^{-1}$] P from areas of
intense agriculture, approximately 10% and 52% of the demand may be satisfied by recovery from urban wastewater and livestock, respectively (62% total), while still excluding any economic dimension.

The results of these optimistic recovery potentials are summarized, per continent, in Table 1. Some continents (i.e. South and North America) show significant disproportionalities in recoverable P from waste vs. phosphorus demand for crop
production. (Virtual) phosphorus trade (e.g. soy bean products) can play an important role in determining these continental budget surpluses and deficits. In the end, however, the total phosphorus budget is only slightly off balance at 109% of the total production potential to demand for the density maps. This global 9% surplus suggests that there is an inherent *overestimation* of the phosphorus production (excretion) or *underestimation* of the demand (agriculture), or that some degree of soil nutrient mining by the crops is considered in the phosphorus requirement values presented in 'Fertilizers and Their
Use' (FAO & IFA, 2000). Another explanation for this disproportionality is that non-agricultural consumers of phosphorus (e.g. medicine and detergents industries) are not considered as actors even though their consumed products are included in

the wastewater discharge figures. The 9% overestimation could suggest a 9% share of these actors in the global phosphorus market. Including these actors would raise the demand and likely close the deficit.

### 3.2. Phosphorus recovery potentials including economic dynamics

Phosphorus quantities, prices, and the distances between the nodes are used to determine market prices at which phosphorus
trade may occur globally. This is used to assess the more realistic, spatially dependant and economically constrained, phosphorus recovery potential. The production costs for phosphorus for the different actors are summarized in fig 3.

In a 2015 market where sustainable products compete with rock-based fertilizers (scenario 2), the model determines that approximately 0.15 [Mt] can be economically recovered, thereby satisfying 0.8% of total agricultural demand. In a market
without rock-based fertilizer products, approximately 7.92 [Mt] can be economically recovered, satisfying 41% of the total agricultural demand. Due to differences in total supply and demand amongst the scenarios, both the market prices and total quantities traded will vary. Optimal trade in a near-future scenario of recovered and mined phosphorus products (scenario 2) occurs at a market price of 2,039 [$ $t^{-1}$]. For a market of only recovered products (scenario 3), where there exists a strong deficit in phosphorus supply for agriculture, optimal trade occurs at much higher prices of 5,700 [$ $t^{-1}$]. The model price and
trade determinations for 2015 and other years are summarized in Table 2.

Associated with these quantities and prices are network trade maps. Although easily created for every year and for every price, only those relate to scenario two and three, for 2015, are presented in figures 5 and 6, respectively. The high recovery potential and close proximity of recovery nodes to agricultural demand nodes makes phosphorus recovery in Asia
particularly competitive in both scenarios. The struvite scaling maintenance and sludge handling cost savings in developed areas (with BNR equipped WWTPs) results in competitive trade in Europe and the United States also. Because compost pellets, which due to their low P-density are far more expensive per tonne phosphorus than other products, the market prices for phosphorus are driven upwards in the 3[rd] scenario. This is a result of the model economics, where the different commodities (struvite vs. compost pellets) are treated as products acting on the same market - discriminated only based on
their phosphorus content. When struvite producers observe consumers buying the relatively expensive (dollars per amount P) compost pellets due to the depletion of struvite suppliers, the initial struvite sellers will adjust their prices upwards forcing consumers to pay more for the same amount of P simply because they can (they are profit driven). The inverse would be true also if agricultural consumers observed cheaper trades occurring among other actors. They would then demand lower prices from their producer or switch producer all together, resulting in lower market prices.

### 3.3.3. Model Validation and Sensitivity Assessment

Due to the hypothetical nature of global phosphorus recovery from wastewater, model performance can only be assessed by comparing the quantities and prices produced by the model for the 'mines only scenario' (scenario 1) with observed and recorded DAP price statistics. These prices, along with the results for the other two (more hypothetical) modelling scenarios,
are presented in fig. 4. The graph of figure 4 shows that the model is able to adequately reproduce DAP price trends. Only the price estimates of 2009, 2012 and 2013 are significantly different to those recorded, as they are 37%, 29% and 31% higher respectively. This may be a delayed response to price stabilization after the inflations of 2008 and 2011. On average,

over all the years, the model predictions show a 17% difference with the observed prices, with maximum difference of 37% in 2009, and a minimum difference of 3% for 2008 and 2015.

Model sensitivity is determined through the *one-at-a-time* (OAT) method. Following this technique, the value of one parameter is adjusted and the model is re-run to evaluate how significant the change in model output is as a result of the parameter change. For each of the individual 31 parameters or input data, we assess how sensitive the model is to a -50% to +50% change in original value. We assess the impact of this change for the model determined i) total phosphorus trade, ii) sustainable phosphorus trade, and iii) optimal market price, for a 2015 market of both recovered and mined products.

The sensitivity analysis shows variable sensitivity to changes in parameter (Table S7 and Figure S3 of Supplementary Materials). Provided that each different year has different input data, these sensitivities will likely vary also depending on the simulation year. For example, in a year where rock phosphate exploitation costs are high (e.g. 2008), the market may be much more dependent on recovered products than in other years. The total phosphorus trade will then likely fluctuate much more with the price of, e.g., magnesium chloride, than determined in the sensitivity analysis conducted for 2005.

There is a high sensitivity of the optimal price and quantity of sustainable trade with variations in transport parameters related to over-seas transport (Table S7). By far the most sensitive parameter is a ship's carry load. A 50% decrease therein, for example, may result in upto 85% increase in phosphorus prices. The second most sensitive is a ships velocity, where a 50% decrease results in 42% higher prices. Unfortunately, there exists high uncertainty also around these parameters as there exist innumerable shipping options (ship loads, -fuel efficiencies, costs, etc) for the transport of phosphorus over seas. Oppositely, the model results show to be insensitive to changes in recovery parameters, suggesting that at global scale the transportation cost of products have a much greater weight in determining the feasibility for trade than the production costs do. Also the relatively small contribution of urban recovered P (maximum 10% of global P-demand, section 3.1) may be a reason why the global P market prices are so insensitive to small changes in the parameters that determine the P-recovery cost.

**4 Discussion**

Many generalizations, assumptions, and simplifications have been made in this study. The lack of consideration for immediate on-site recycling, trans-Atlantic movement, assumptions of free trade, and other economic simplifications, are among those possibly contributing to errors in market price determination and patterns of trade flow. Furthermore, external costs such as those associated to environmental impacts (e.g. $CO_2$ emissions and energy requirements) for the various fertilizer production/recovery chains has been neglected to keep the economic analysis simple. Including these costs would likely significantly improve the favourability of phosphorus recovery from wastewater considering the reduced transportation distances and relatively $CO_2$ neutral recovery technologies (Molinos-Senante et al., 2011). Many of the assumptions of this study are partially justified by the global and explorative nature of this investigation on potentials. An overview of all significant assumptions and their possible implications are presented in Table 3.

The lack of studies on the global *economic potentials* for recovery and phosphorus trade patterns at subnational resolution, inhibits comparison of this study's primary results. The model results on *total potentials* and struvite pricing, however, are

well aligned to those of other studies. Above we determined that in 2015, 3.7 [Mt a$^{-1}$] of phosphorus is discharged into wastewater, satisfying 20% of the reported 18.52 [Mt a$^{-1}$] agricultural demand (Heffer and Prud'homme, 2016), and 19% of determined 19.51 [Mt a$^{-1}$] agricultural demand.

**Smil (2000)** found 3 [Mt a$^{-1}$] potentially recoverable, which would account for 20-25% of the global agricultural demand. Extrapolating Smil's (2000) figure proportionally with a population growth of 1.22% per year would result in a potential urban production of 3.6 [Mt a$^{-1}$] in 2015, a less than 3% difference from the model determined potential. **Mihelcic et al. (2011),** through a study on diets and phosphorus excretion, concludes that the phosphorus excretion rates per individual can vary as much as from 0.18 P [kg a$^{-1}$] in the Democratic People's Republic of Congo, to 0.73 P

[kg a$^{-1}$] in Israel. This confirms that our 'Western' approximation for phosphorus excretion of 0.77 P [kg a$^{-1}$] is on the global high end. For 2009, nonetheless, Mihelcic et al. (2011) determines that 3.4 P [Mt a$^{-1}$] of human waste produced could account for 22% of the 15 [Mt a$^{-1}$] of global phosphorus demand.

      **Van Drecht et al., (2009)** considers variability in access to sanitation, livings standards, and other population relevant variables to determine a phosphorus discharge of 1.3-3.1 [Mt a$^{-1}$] to wastewater systems in the period 2000 to 2050.

Similarly, **Morée et al., (2013)** determines a P discharge of 0.2 to 1.0 [Mt a$^{-1}$] from urban wastewater over the period 1950 to 2000. These are lower estimates than the ones produced in this study, suggesting that our assumptions of i) everyone being connected to some form of sanitary infrastructure and ii) everyone discharging phosphorus according to western throughput figures, is unrealistically optimistic even for near future scenarios. However, Morée et al., (2013) also determines that, over that same period, 0.08 [Mt a$^{-1}$] of P was recycled back to agriculture, which is lower,

but in range to our estimated economic recovery potential of 0.13 [Mt a$^{-1}$] for 2015.

      **Bouwman et al., (2013),** also using slaughter weights, determined that 17 [Mt a$^{-1}$] of P is produced by livestock in 2000, matching closely the model determined 17.11 [Mt a$^{-1}$]. **Bouwman et al. (2011)** notes the potential for recovery in industrialised countries, which is in line with the general trade patterns presented in fig. 5.

      **Koppelaar and Weikard (2013)** estimate 4.2 [Mt a$^{-1}$] P domestic discharge, which is 25% of their 16.7 P [Mt a$^{-1}$]

agricultural demand. They also estimated total domestic animal manure production of 28.3 [Mt a$^{-1}$] P. This is distinctly higher than this study's 17.11 [Mt/year] P from livestock. This is likely in part due to the fact that they account for a much larger variety of livestock types than the cattle, poultry, and swine considered in this study.

The model estimates struvite *production costs* ranging from 0 to 670 [$ t$^{-1}$] variable with the nature and location of the

30 recovery site (fig. 3; converted prices[2]). Phosphorus *market prices* for a market that also offers rock phosphate products range from 273 to 391 [$ t$^{-1}$] over the different years (fig. 4; converted prices). The supply deficit resulting from a scenario without rock phosphate products, drives these sale prices upwards to a range of 570 to 955 [$ t$^{-1}$]. These costs and prices, like the recovery potentials, are difficult to compare provided no found study has evaluated the prices for economic recovery potential at global scale. Instead, there exist case-studies on the feasibility for phosphorus recovery at specific sites.

35

---

[2] Prices in this study are mostly represented as U.S. dollars per tonne phosphorus. For comparison purposes, we can convert these to DAP prices through multiplication with DAP's P density (24%), or to struvite prices through multiplication with struvite's P density (14%).

**Ueno and Fujii (2001)** observed that struvite obtained from wastewater in Japan is sold to fertilizer companies at rates of 300 [$ t$^{-1}$]. A market study by **Münch and Barr (2001)** revealed that struvite can be sold in Australia for between 220 and 370 [$ t$^{-1}$]. **Shu et al. (2006)**, however, estimated that the market price of struvite is around 550 [$ t$^{-1}$]. Based on fertilizer market estimation, **Dockhorn (2009)** estimated far higher prices than those mentioned before, and values recovered struvite at 900 [$ t$^{-1}$]. Dockhorn's high recovered product prices are approached in the model by the 2015 struvite market price, 955 [$ t$^{-1}$], for a market scenario with no rock phosphate competition and a severe P supply deficit. It appears that the model determined price range for global struvite production covers the spectrum of different production costs as determined in various different other studies.

In this investigation we consider struvite precipitation as the primary means to recover phosphorus from wastewater. In reality, there are many other recovery technologies that also offer high recovery rates (Cordell et al., 2011; Egle et al., 2016). It would be an interesting follow-up study to adapt and run the model for different phosphorus recovery technologies and visualize changes in trade patterns for different phosphorus products.

**5 Conclusion**

Despite the simplifying assumptions, the model developed in this study generates realistic trade networks for different phosphorus supply scenarios, for different prices, at a subnational resolution. However, the credibility of model outputs is only supported by an accurate simulation of DAP prices because data and/or other studies on the purely hypothetical nature of global trade in sustainably recovered P, are lacking. Nevertheless, the model sets a basis that provides some general indication of the spatially dependant recovery feasibility of phosphorus from wastewater. It is furthermore able to provide this indication for potentially any recovery technology for which there exists adequate economic data.

Model results reveal a relatively minor potential of economically profitable, struvite fertilizer production from wastewater. This recovery thus appears to offer a limited contribution to resolving the global phosphorus issues of the 21$^{st}$ century. Nevertheless, at a more local scale , this recovery offers wastewater treatment plants the opportunity to contribute to creating sustainable communities and protecting the environment, while reducing their own operational cost. This potential exists foremost in the highly populated cities of developed countries.

Although recognizing that there is no single solution to solving phosphorus pollution and insecurity issues (Cordell and White, 2011), recovering phosphorus from all waste sources may come to provide a greater contribution as populations grow and urbanize, technologies develop, and the economically extractable phosphorus reserves deplete. For this reason, it is essential to determine how the widespread implementation of recovery technologies impacts phosphorus market dynamics. Only then can we stimulate and regulate its recovery in such a way that maximum benefits are achieved for both the environment and the urban community, as well as the livestock and agricultural sectors.

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

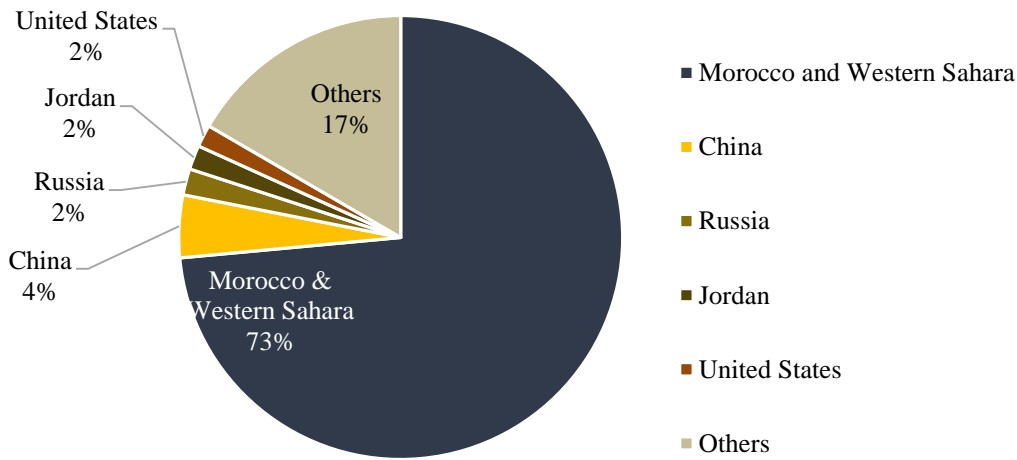

**Figure 1. Estimated global phosphorus reserve distribution. The vast majority (73%) of estimated natural reserves lie in Moroccan and West-Saharan territories (USGS, 2017).**

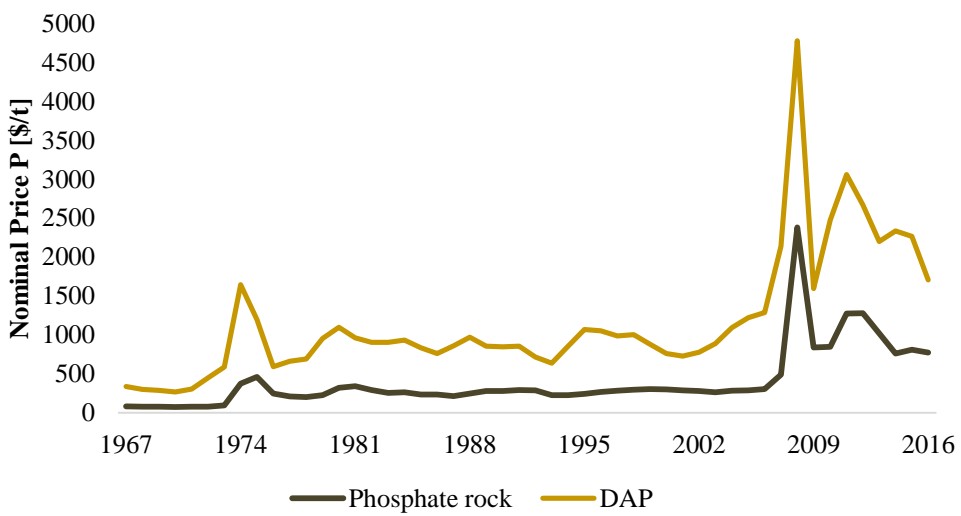

**Figure 2. Nominal Phosphate rock and DAP price trends (IndexMundi, 2017). The rock phosphate and DAP price trend can be characterized as gradually increasing and vulnerable to market dynamics (i.e. the 2007 global economic recession).**

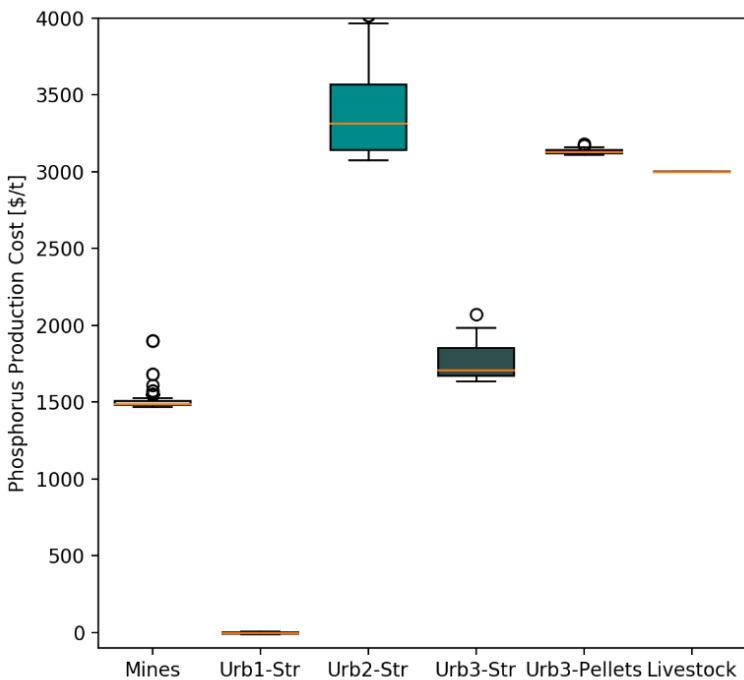

**Figure 3. Minimum production costs per ton of recycled P for supply nodes. Urb1-Str shows the prices for struvite from WWTP in countries with highly developed sanitation systems (BNR), where the struvite is offered for free (excl. transport costs) because of the maintenance and sludge handling savings by controlled struvite precipitation; Urb2-Str, from nations with slightly less developed sanitation systems; Urb3-Str, of source-separated urine collection and struvite precipitation in nations with underdeveloped sanitation systems; Urb3-Pellets, the price per tonne P (0.1%) in urban compost pellets; and Livestock shows the price per tonne P in livestock pellets (0.1% P).**

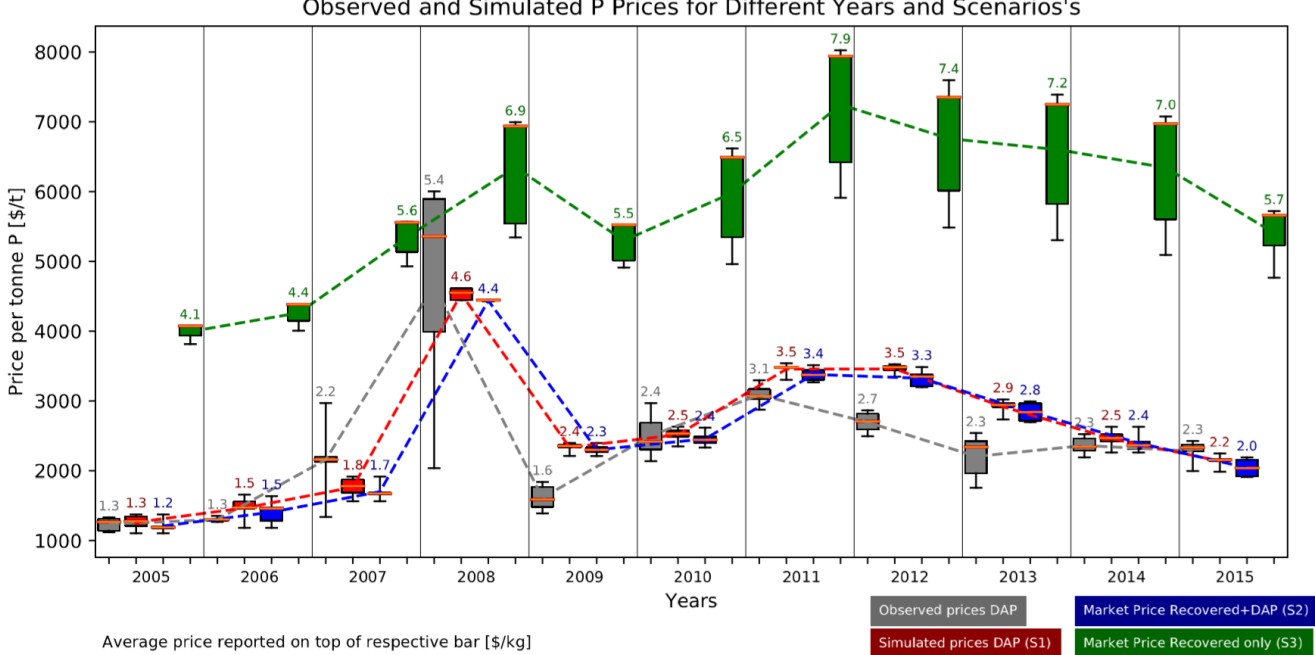

**Figure 4. Plausible phosphorus price ranges at which trade can occur at global scale for different years: observed prices (grey) versus modelled scenarios (Sc. 1, pink; Sc. 2, blue; Sc. 3, green). The ends of the whiskers on the grey boxes for the observed prices represent the maximum and minimum prices for DAP that year, the box itself shows the upper and lower quartiles for that data and the orange line indicates the median. For the modelled scenario the whisker-ends show the maximum and minimum prices as approximated by the model, the box indicates the most likely price range, and the orange line marks the most probable price given supply-demand ratio's. Noticeable is that the model price determinations for the current mines only scenario are in close proximity with observed price ranges. Scenario 2 and 3 are hypothetical scenarios and therefore do not have a realistic counterpart dataset that can be used for comparison. Nevertheless, they show predictable and realistic behaviour.**

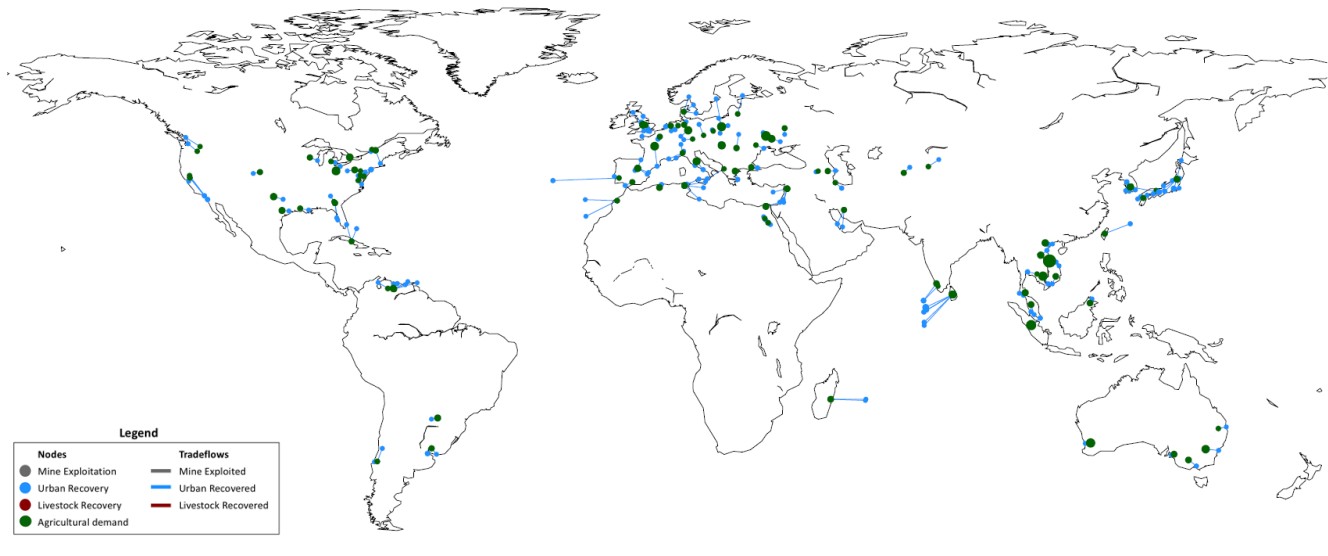

**Figure 5. Phosphorus trade network for trade in both conventional and recycled P (scenario 2) but showing only sustainable trade flows. Optimal trade occurs at a phosphorus market price of 2,039 [$ t⁻¹] with 16.81 [Mt] being traded in total, of which 0.15 [Mt] is traded sustainably (0.8% of total demand).**

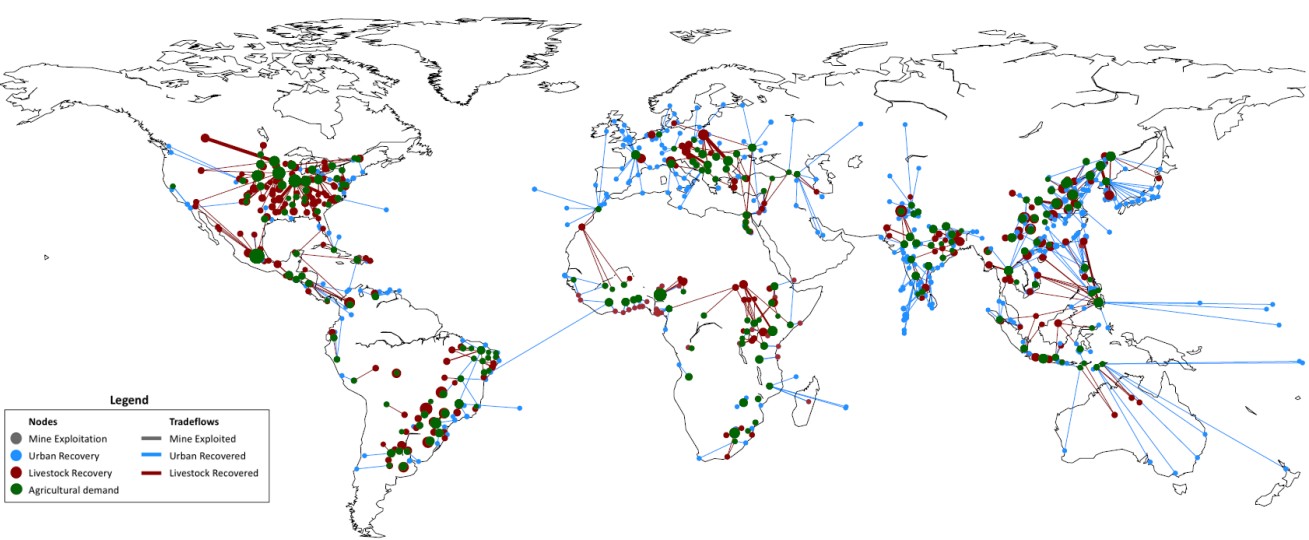

**Figure 6. Phosphorus trade network for trade in recycled P only (scenario 3) at determined phosphorus market prices of 5,700 [$ t⁻¹] with 7.92 [Mt] being traded. All of this trade is sustainable but is only able to accommodate 41% of demand.**

**Continental Phosphorus Budgets**

| | Livestock Production | | Human Production | | Agricultural Demand | |
|---|---|---|---|---|---|---|
| | Total [Mt P] | Major Nodes [Mt P] | Total [Mt P] | Major Nodes [Mt P] | Total [Mt P] | Major Nodes [Mt P] |
| **Asia** | 5.85 | - | 2.29 | - | 9.49 | - |
| **North America** | 3.10 | - | 0.31 | - | 3.30 | - |
| **Europe** | 2.39 | - | 0.40 | - | 1.79 | - |
| **Africa** | 1.72 | - | 0.53 | - | 2.51 | - |
| **South America** | 3.89 | - | 0.22 | - | 2.17 | - |
| **Oceania** | 0.05 | - | 0.00 | - | 0.00 | - |
| **Australia** | 0.28 | - | 0.01 | - | 0.24 | - |
| **World** | 17.11 | 10.47 | 3.75 | 1.73 | 19.52 | 16.81 |

**Table 1. Approximate continental phosphorus budgets associated to maps. The 'Total' values represent an estimation of recovering all phosphorus on the continent, while the 'Major Nodes' values represents that which can be recovered at sites of high potential only.**

## Model Trade and Price Determinations

| | | Scenario 1 | Scenario 2 | Scenario 3 | Units |
|---|---|---|---|---|---|
| **2005** | Maximum Traded | 15.01 | 15.01 | 5.59 | [mt] |
| | Sustainably Traded | 0 | 0.12 | 5.59 | [mt] |
| | % of total demand met | 100 | 100 | 37 | [-] |
| | % of total demand met sustainably | 0 | 0.80 | 37 | [-] |
| | Optimal Price | 1276 | 1189 | 4000 | [$ t$^{-1}$] P |
| | Min price for 90% of max. trade | 1000 | 1000 | 3700 | [$ t$^{-1}$] P |
| | Max price for 90% of maximum trade | 21000 | 2100 | 4000 | [$ t$^{-1}$] P |
| **2006** | Maximum Traded | 15.46 | 15.46 | 5.89 | [mt] |
| | Sustainably Traded | 0 | 0.12 | 5.89 | [mt] |
| | % of total demand met | 100 | 100 | 38 | [-] |
| | % of total demand met sustainably | 0 | 0.78 | 38 | [-] |
| | Optimal Price | 1559 | 1460 | 4300 | [$ t$^{-1}$] P |
| | Min price for 90% of max. trade | 1100 | 1100 | 3800 | [$ t$^{-1}$] P |
| | Max price for 90% of maximum trade | 2300 | 2300 | 4300 | [$ t$^{-1}$] P |
| **2007** | Maximum Traded | 15.91 | 15.91 | 6.59 | [mt] |
| | Sustainably Traded | 0 | 0.12 | 6.59 | [mt] |
| | % of total demand met | 100 | 100 | 41 | [-] |
| | % of total demand met sustainably | 0 | 0.75 | 41 | [-] |
| | Optimal Price | 1780 | 1674 | 5500 | [$ t$^{-1}$] P |
| | Min price for 90% of max. trade | 1500 | 1500 | 4000 | [$ t$^{-1}$] P |
| | Max price for 90% of maximum trade | 2900 | 2900 | 5500 | [$ t$^{-1}$] P |
| **2008** | Maximum Traded | 10.03 | 10.03 | 7.12 | [mt] |
| | Sustainably Traded | 0 | 6.91 | 7.12 | [mt] |
| | % of total demand met | 62.69 | 62.69 | 45 | [-] |
| | % of total demand met sustainably | 0 | 43.19 | 45 | [-] |
| | Optimal Price | 4551 | 4445 | 6900 | [$ t$^{-1}$] P |
| | Min price for 90% of max. trade | 4700 | 3500 | 3500 | [$ t$^{-1}$] P |
| | Max price for 90% of maximum trade | 6900 | 6900 | 6900 | [$ t$^{-1}$] P |
| **2009** | Maximum Traded | 16.81 | 16.81 | 6.82 | [mt] |
| | Sustainably Traded | 0 | 0.16 | 6.82 | [mt] |
| | % of total demand met | 100.00 | 100.00 | 41 | [-] |
| | % of total demand met sustainably | 0 | 0.95 | 41 | [-] |
| | Optimal Price | 2355 | 2299 | 5500 | [$ t$^{-1}$] P |
| | Min price for 90% of max. trade | 2100 | 2100 | 4000 | [$ t$^{-1}$] P |
| | Max price for 90% of maximum trade | 2900 | 2900 | 5500 | [$ t$^{-1}$] P |
| **2010** | Maximum Traded | 17.26 | 17.26 | 7.13 | [mt] |
| | Sustainably Traded | 0 | 0.16 | 7.12 | [mt] |
| | % of total demand met | 100.00 | 100.00 | 41 | [-] |
| | % of total demand met sustainably | 0 | 0.93 | 41 | [-] |
| | Optimal Price | 2534 | 2444 | 6618 | [$ t$^{-1}$] P |
| | Min price for 90% of max. trade | 2200 | 2200 | 3400 | [$ t$^{-1}$] P |

| | | | | | |
|---|---|---|---|---|---|
| | Max price for 90% of maximum trade | 3400 | 3400 | 6600 | [$ t$^{-1}$] P |
| **2011** | Maximum Traded | 17.71 | 17.71 | 8.16 | [mt] |
| | Sustainably Traded | 0 | 0.6 | 8.16 | [mt] |
| | % of total demand met | 100.00 | 100.00 | 46 | [-] |
| | % of total demand met sustainably | 0 | 3.39 | 46 | [-] |
| | Optimal Price | 3481 | 3374 | 8000 | [$ t$^{-1}$] P |
| | Min price for 90% of max. trade | 3100 | 3100 | 3600 | [$ t$^{-1}$] P |
| | Max price for 90% of maximum trade | 4200 | 4200 | 8000 | [$ t$^{-1}$] P |
| **2012** | Maximum Traded | 18.14 | 18.16 | 7.9 | [mt] |
| | Sustainably Traded | 0 | 0.59 | 7.12 | [mt] |
| | % of total demand met | 99.89 | 100.00 | 44 | [-] |
| | % of total demand met sustainably | 0 | 3.25 | 44 | [-] |
| | Optimal Price | 3484 | 3348 | 7596 | [$ t$^{-1}$] P |
| | Min price for 90% of max. trade | 3100 | 3000 | 3500 | [$ t$^{-1}$] P |
| | Max price for 90% of maximum trade | 3900 | 3900 | 7600 | [$ t$^{-1}$] P |
| **2013** | Maximum Traded | 18.61 | 18.61 | 8.01 | [mt] |
| | Sustainably Traded | 0 | 0.2 | 8.01 | [mt] |
| | % of total demand met | 100.00 | 100.00 | 43 | [-] |
| | % of total demand met sustainably | 0 | 1.07 | 43 | [-] |
| | Optimal Price | 2943 | 2840 | 7389 | [$ t$^{-1}$] P |
| | Min price for 90% of max. trade | 2500 | 2500 | 3500 | [$ t$^{-1}$] P |
| | Max price for 90% of maximum trade | 3800 | 3800 | 7400 | [$ t$^{-1}$] P |
| **2014** | Maximum Traded | 19.06 | 19.06 | 8 | [mt] |
| | Sustainably Traded | 0 | 0.15 | 8 | [mt] |
| | % of total demand met | 100.00 | 100.00 | 42 | [-] |
| | % of total demand met sustainably | 0 | 0.79 | 42 | [-] |
| | Optimal Price | 2469 | 2365 | 7000 | [$ t$^{-1}$] P |
| | Min price for 90% of max. trade | 2000 | 2000 | 3500 | [$ t$^{-1}$] P |
| | Max price for 90% of maximum trade | 3700 | 3700 | 7000 | [$ t$^{-1}$] P |
| **2015** | Maximum Traded | 19.51 | 19.51 | 7.92 | [mt] |
| | Sustainably Traded | 0 | 0.15 | 7.92 | [mt] |
| | % of total demand met | 100.00 | 100.00 | 41 | [-] |
| | % of total demand met sustainably | 0 | 0.77 | 41 | [-] |
| | Optimal Price | 2155 | 2039 | 5700 | [$ t$^{-1}$] P |
| | Min price for 90% of max. trade | 1900 | 1900 | 4000 | [$ t$^{-1}$] P |
| | Max price for 90% of maximum trade | 3000 | 3000 | 5700 | [$ t$^{-1}$] P |

**Table 2. Price ranges and amounts traded per scenario for all years.**

## Overview of Assumptions

| Methods | Assumption | In Reality | Implication |
|---|---|---|---|
| **Production and demand estimates.** | Crop phosphorus requirement only varies with crop type and water-constrained yield. | - A particular crop yield may be constrained by other factors than water availability (e.g. soil acidity levels, micronutrient levels, management practices, etc.), also influencing the actual phosphorus demand.<br>- Crop phosphorus requirements depend also on soil dynamics processes determining leaching rates and sorption. | The phosphorus demand quantities will be spatially more variable and are likely overestimated. |
| | Domestic wastewater composition is globally homogeneous, and its quantity only varies with population density | - Wastewater composition and amount vary amongst different populations. Both quantity and quality factors of wastewater depend on the amount and nature of connected industries and lifestyle characteristics of the people connected (e.g. diets and detergents use), the regional climate, as well as whether or not the sewerage network is a combined or separate system, etc. | Recovery efficiencies will likely be lower, provided the current assumption of western lifestyles, globally. |
| | We can approximate the type of wastewater treatment practiced in a country based on data that shows the percent of urban population that have access to sanitary facilities. | - The type of wastewater treatment at a node varies with many different socio-economic (and natural) parameters that it can hardly be approximated for using a single dataset. | Even in the near future, few people in developing countries will have access to sanitation resulting in an even lower recovery efficiency of phosphorus from developing regions than is currently predicted. |
| | Phosphorus recovery efficiency is determined solely by the recovery technology. | - Phosphorus recovery efficiency varies not just per technology, but also with the wastewater composition. | Node specific phosphorus production potentials can potentially be higher or lower than is currently determined. |
| | Phosphorus throughput per individual is globally homogeneous | - Phosphorus excretion rates vary enormously depending on age, diet, and gender of the individual.<br>- Phosphorus discharge relates also to population lifestyles which determine the diet (as described above) but also detergent use. | Node specific phosphorus production potential is lower in developing countries than is currently determined. |
| | Assume container ship as sea transport mode | - Bulk trade occurs in bulk carriers. Depending on how finished the product is (e.g. bagged and sealed), it may be transported by container also, | Sea transport may be significantly cheaper than is current the case. |
| | Fertilizer maximum bid price depends on crop | - Farmers growing the same crop, requiring the same amount of fertilizer, may have different maximum prices for P fertilizers depending on other factors (e.g. experience, subsidies, crop quality and client for which the crop is grown, hence different crop values) | Recovery feasibility will be higher or lower depending on the profitability and experience in agriculture in an area. |
| **Modelling trade** | All phosphorus goes to the international fertilizer market. | - Most phosphorus is recycled locally, applied on nearby agricultural soils directly as manure or as treated wastewater sludge (ash). | Phosphorus recovery potentials are underestimated given the disregard for immediate local use. Phosphorus production from mines |

| | | |
|---|---|---|
| | - Some is distributed amongst pharmaceutical and detergent industries | is overestimated, as not all is used for fertilizer production. |
| Free trade | - Trade does not stand separate from politics. Embargo's, trade sanctions, or trade tariffs can influence the pattern and amount of global phosphorus trade. | Trade patterns would look different provided international politics. |
| Non-preferential trade | - Trade is not purely rational. Some countries/actors may be more or less likely to trade with each other depending on historical and current relations. | Trade patterns would look different provided international relations. |
| Two actor trades | - Phosphorus extraction from phosphate rock and its processing into artificial fertilizers may not occur at the same site, nor by the same actor. Often it involves many more parties. One for exploitation, for manufactory, for logistics, etc. | Trade patterns are simplified, showing only the path from site of initial production to site of final demand. |
| Transportation distances are calculated as straight lines as opposed to following existing infrastructure. | - Phosphorus transport in reality follows existing shipping routes and road infrastructure. | The model likely underestimates the transportation cost of moving phosphorus from supply to demand areas. However, it does so consistently for both recovered as well as mined products. This nevertheless reduces recovery potential estimates. |
| No transatlantic trade due planar projection of earth. | - International trade is not restricted by cartographic boundaries. | Few implications for the simulated years as trade in phosphorus between the America's and Asia is unlikely to occur due to the greater distances and relatively balanced continental phosphorus budget of North America. |
| Near-Future and Far Future scenarios do not consider development of technology | - Existing technologies are likely to become cheaper and more efficient in the future, while new technologies may also be developed | Recovery will rates will likely be higher in the future, thereby also reducing the price of phosphorus. However, provided the uncertainty around making technological development predictions, we decided to exclude this factor from our analysis. |

**Table 3. An overview of the assumptions made in this study and their possible implications on the results.**

