# Peer review of "Global Phosphorus Recovery from Wastewater for Agricultural Reuse"

_Hydrology and Earth System Sciences, 2018_

## Referee Comment (RC1) · Anonymous Referee #1 · 27 Apr 2018

The manuscript tries to identify potential phosphorus source and demand areas and connect them through market price. Phosphorus is an important input in crop production with potential for causing water pollution if not managed properly. The topic is very interesting for wider audience. The manuscript is well written with a good analysis of the different sources and supply areas. However, some of the assumptions and methods employed need further clarification before the manuscript got accepted.

**1. Manure is an important source of P. However, it is not clear from the text what is included in the manure source. (a) How did you account for the manure that falls on pasture during grazing and the part that falls on roadside? Depending on the production system, part of cattle manure may fall in the pasture and won't easily be collected. Please see Bowuman et al. (2009, 2013) and the associated detailed description**

Creative Commons BY license logo

of the model used (http://www.pbl.nl/en/publications/2011/exploring-global-changes-in-nitrogen-and-phosphorus-cycles-in-agriculture-induced-by-livestock-production-over ). Please provide detailed description of the method you followed. (b) Manure production per head is dependent on the volume of feed animals consume. It is not right to assume all animals in both developed and developing countries will generate same amount of manure per head as given in Table S1. Sheldrick et al. (2003) provide the manure production per animal as a function of slaughter weight. The same approach was followed by Liu et al. (2010), and Mekonnen and Hoekstra (2017). I suggest you differentiate the amount of manure per head if possible per country if not at least per economic development.

**2. Equation (2) is confusing and no detail on the input provided. The equation by Doorenbos and Kassam (1979) or Steduto et al. (2012) is specified for the growing period:**

so I am not sure the need for the additional parameters (Tg, AH, and C). If you are aggregating the actual ET over the growing season, why do you need a correction factor? I would expect both the crop water requirement and the actual ET to be over the growing season and over the cropping area. If that is the case why put Tg and Ah on the nominator but not in the denominator? The other question I have is how did you estimate the crop evapotranspiration (ET) and the crop water requirement (CWR) per grid per crop? There is no detail where and how you got the data. What value did you assume for the maximum yield (Ym)? Does it vary per country or climate region or one value for the global? If you have to estimate the yield please do it properly. Otherwise, one suggestion is to use the yield from other sources such as Monfreda et al. (2008).

**3. Determination of the node: sorry if I have missed it but it is not clear from the manscript how the demand and supply node were determined. How did you define the location of the node? What parameters did you use? Please clarify!**

**4. I was wondering if you have accounted for the legacy P or the residual P that**

accumulates in the soil in determining the demand nodes. The residual P can be taken up by crops so it can be considered one source of P.

**5. The result are dependent on the number of assumptions and the data used. Therefore, I expect a range of values that describe the uncertainties involved in the modeling. I suggest you provide uncertainty analysis covering the major uncertainties around the input parameters and the model.**

**6. Table 1 is not clear. Why the values for Livestock Raster vs Livestock node and Human raster vs urban node are different? Why do you need to provide two estimate of P for livestock and the human? Please add explanation of the values and why they are different. I also not able to find Figure S2a, S2b, and S2c. Do you mean Figure S1 instead of S2?**

**7. It will enhance the reliability of your result if you compare the result with earlier estimates such as Bouwman et al. (2013) for manure; Van Drecht et al. (2009) and Moree et al. (2013) for domestic**

Reference:

Bouwman, L., K. K. Goldewijk, K. W. Van Der Hoek, A. H. W. Beusen, D. P. Van Vuuren, J. Willems, M. C. Rufino, and E. Stehfest (2013), Exploring global changes in nitrogen and phosphorus cycles in agriculture induced by livestock production over the 1900–2050 period, Proc. Natl. Acad. Sci. USA, 110(52), 20882-20887, doi:10.1073/pnas.1012878108.

Bouwman, A. F., A. H. W. Beusen, and G. Billen (2009), Human alteration of the global nitrogen and phosphorus soil balances for the period 1970-2050, Global Biogeochemical Cycles, 23, GB0A04, doi:10.1029/2009gb003576.

Liu, J., L. You, M. Amini, M. Obersteiner, M. Herrero, A. J. B. Zehnder, and H. Yang (2010), A high-resolution assessment on global nitrogen flows in cropland, Proceedings of the National Academy of Sciences, 107(17), 8035-8040,

doi:10.1073/pnas.0913658107.

Mekonnen, M.M. & Hoekstra, A.Y. (2017) Global anthropogenic phosphorus loads to fresh water and associated grey water footprints and water pollution levels: A high-resolution global study, Water Resources Research, 54(1): 345–358.

Monfreda, C., N. Ramankutty, and J. A. Foley (2008), Farming the planet: 2. Geographic distribution of crop areas, yields, physiological types, and net primary production in the year 2000, Global Biogeochemical Cycles, 22(1), GB1022, doi:10.1029/2007gb002947. (data available at- http://www.earthstat.org/data-download/)

Morée, A. L., A. H. W. Beusen, A. F. Bouwman, and W. J. Willems (2013), Exploring global nitrogen and phosphorus flows in urban wastes during the twentieth century, Global Biogeochemical Cycles, 27(3), 836-846, doi:10.1002/gbc.20072.

Van Drecht, G., A. F. Bouwman, J. Harrison, and J. M. Knoop (2009), Global nitrogen and phosphate in urban wastewater for the period 1970 to 2050, Global Biogeochemical Cycles, 23, GB0A03, doi:10.1029/2009gb003458.
* * *

---

## Referee Comment (RC2) · Z. Yuan (Referee) · 30 Apr 2018

Phosphorus is a critical nutrient for livings on the Earth and its sustainable supply is very important for food security. So the recovery of phosphorus from agricultural wastes is an important way to approach sustainable phosphorus supply. This manuscript talks about the phosphorus recovery potential of agricultural wastes and is important for sustain phosphorus supply. However, some critical issues need to be considered so as to make it more robust. (1) The manuscript needs to define the phosphorus wastes in section 1.1 so as to help readers understand what you concern about. in this section 1.1, the manuscript tries to talk about the importance of recovering phosphorus wastes into resources, especially the nutrients from rural and urban wastewater? However, all phosphorus wastes look like those in rural and urban

wastewater? Section 2, Identifying where in the world phosphorus-laden wastewaters confirms that the phosphorus wastes are those phosphorus in wastewater? The scope of phosphorus is critical important to understand the methods and results of the manuscript and so please clarify it clearly.

(2) I am confused about the method of determining global market price of phosphorus. The phosphate rock price changes with consideration of phosphate rock supply and demand as shown by authors, as well as other factors such as transportation distance and political issues. Phosphorus chemical products will change in the same way. What you mean the price? it is the price of phosphate rock, phosphorus chemical products (if so, what kind of product?), or the recovered fine phosphorus? or even phosphorus wastes? please explain carefully.

(3) With regards to formula (1) what you mean the phosphorus throughout rate? the direct phosphorus inputs? outputs? or direct and indirect phosphorus inputs or outputs? Furthermore, recovery efficiency depends on the treatment technologies and collected rates (maybe more factors), which differ a lot for urine and excreta. I suggest authors define the production density clearly and explain the formula carefully.

(4) With the phosphorus demand density, it is estimated without considering the phosphorus in the soil? What is the demand density? the new input (balance of theoretic demand and content in soil), or the total theoretic demand of crops? not include the demand of animals? please explain and clarify.

(5) Again, in the section 2.2, it looks like phosphorus content only in wastewater and not include those in solid wastes? If so what about the phosphorus in solid wastes? and in most cases, they are transferred place by place. For example, in animal breeding plants, some collect them together and some separate them, which will change your results completely and so please clarify it clearly.

(6) I am totally confused about the trade model, to my best knowledge, the phosphorus wastes are more suitable for recycling locally and can not be trade in large scale because it is cheap and the cost will be higher if the transportation distance increases. So it will be more reasonable to recycle phosphorus wastes locally, even they can be produced organic fertilizer. Furthermore, the phosphorus deficit can be met with trade of phosphate rock, phosphorus chemical products, or crops, and different product trade will change the model totally. Please explain the difference and what and how they affect the results, especially for the quantification of trade flows.

(7)the issues above will change the results totally and so if these issues were solved, please analyze the results completely.

(8)Figure can be deleted from the manuscript considering it is the basic data. Figure 2 is confusing and what do you mean with phosphate rock? the production? the use or consumption? theoretic demand? or supply capacity? I also suggest authors move the figure 3 and figure 4 to supporting information.

(9) English expressions need to be polished so as to make it more understandable with scientific basis.

---

## Author Comment (AC1) · 30 Apr 2018

That authors would like to thank the anonymous referee for the helpful feedback. The referee has shared very valid concerns that we hope to address in a revised version of the article. In the below, we would like to share our thoughts with regards to the comments made and present how we would like to resolve the issues identified.

1. The referee identifies a lack in detailed explanation in the text on how we determine the P coming from manure. The referee finds it unclear: (a) how we account for manure that falls on pasture during grazing and the part that falls on roadside and; (b) how we determine manure production.

In the study, we identify livestock density distributions using geospatial data (FAO gridded livestock of the world (GLW)). We then attach a single phosphorus throughput estimate based on literature to each individual in each livestock group (cattle, poultry, swine). With regards to manure accumulating in pastures and on roadsides (a), it is assumed that only half of the global excreted phosphorus from cattle is recoverable due to different pastured-stabled periods (pastured for half a year resulting in loss of phosphorus to pastures, and stabled for half a year during which phosphorus can be recovered in pits). This methodology will be emphasized in the new review. With regards to determining global manure production (b), we acknowledge that we had difficulty attaching a phosphorus production potential to each livestock head for the reasons mentioned by the referee (i.e. varies per volume of feed consumed, spatial data that is not available to our knowledge). In fact, the amount of phosphorus excreted will not only vary with the volume of feed consumed, but also vary per quality of feed, the animal species, race, gender, and the animals age. Slaughterhouse weight appears to be a logical index for these parameters and we regret that we did not think of using this ourselves. We will therefore integrate this in the revised version of the article, and would like to thank the referee for providing this particular insight.

2. The referee finds equation 2 confusing and remarks that no detail on the input is provided. The referee also questions the introduction of the additional parameters (Tg, AH, and C) into the equation.

As the referee remarks, the equation by Doorenbos and Kassam (1979) or Steduto et al. (2012) is specified for a single growing period. The ET used should then coincide with this growing period. However, the growing period starts and ends on different dates depending on geographical location (mostly latitude) of the cropped area, and therefore so will the ET. The crop harvested area maps, meanwhile, are yearly and therefore do not disseminate as to when to growing season starts or ends in different parts of the world. Attaching a relevant ET to each area thus becomes complicated as European corn will grow in European summers, while in Latin America it will likely grow during European winters – meanwhile the crop harvest maps only show that both

regions grow corn that year. As such, without manipulating the data, only the growing duration (Tg) can be assumed to be roughly similar for the crops in both areas. To still make an estimate of the potential yield, we disregard for different starts of growing seasons, and calculate ET over the entire year. We then assume that a proportion equal to the growing season duration (Tg) of that year's ET is available to the crop (Tg/365). Thus we ignore ET seasonality, i.e. assume it to be constant over the duration of a year. Then the ET attributed to growing periods are corrected for the area harvested (AH), again assuming that ET is independent of landcover type. Due to these induced assumptions, some parts of the world achieve higher than optimum yields. To correct for this, C was introduced that scales everything above optimum, back down to optimum. Other parameters (e.g. water requirements and crop specific data) were taken from FAO and are assumed globally homogenous. Since estimating the potential yield is only a small (but important) component of this broad scoped study, we agree that integrating Monfreda's (2008) potential yield data offers a better, more credible alternative.

3. The referee finds it unclear from the manuscript how the demand and supply nodes are defined, and more specifically how we determine their location.

In the study, the locations for demand and supply nodes are determined using geospatial data. For unsustainable supply nodes, the mine locations are taken directly from a 2002 USGS vector dataset. For sustainable supply nodes (wastewater accumulation sites), we simply multiply population density rasters with phosphorus throughput statistics (discussed for livestock in point 1) to acquire phosphorus production density rasters. Large, connected areas of high phosphorus production potential are then converted to nodes using GIS tools. To ensure that, for example, the highly populated German Ruhr region is not merged together in a node that represents most of the Netherlands, the raster zones are separated by administrative boundaries. The exact location of the node is in the middle of each separated, high potential density area. This aspect of the methodology will be more explicitly explained in the updated version

of the manuscript.

4. The referee wonders if we account for the legacy P or the residual P that accumulates in the soil.

Although it was considered, we decided not to complicate this study further by including changes in soil stored P. Instead, we treat phosphorus like a hydrological bucket model, where we assume the year to year change in soil stored phosphorus to be 0. The yearly demand is then equal to the yearly phosphorus requirements of crops to meet corresponding water-constrained yield. To emphasize this, we will create an comprehensive overview of all the assumptions in this study

5. The referee remarks the absence of an uncertainty analysis with regards to modeling.

The lack of an uncertainty analysis is concern shared with the authors. We refrained from doing this analysis provided the many assumptions, parameters and variables included in the model which we felt illustrated qualitatively quite well the uncertainty around the results. Recognizing again that doing a quantitative uncertainty analysis is standard procedure for any modeling investigation, we will explore which form of uncertainty analysis is most appropriate for this study.

6. The referee remarks that Table 1 is not clear, and wonders why the values for Livestock Raster vs Livestock node and Human raster vs urban node are different? He also remarks that he is unable to find Figure S2a, S2b, and S2c.

Table 1 is unclear because of improper definitions in the text of the differences between the 'raster' and 'node' values. The difference between 'raster' and 'node' values exists to account for differences in definition of 'recovery potential'. The raster value represents the how much phosphorus could become available if we recover phosphorus everywhere (across the entire raster), while the node value represents only that amount if we recover from high production sites only. This is to say that not all raster

values are translated into nodes. Low phosphorus production densities are systematically excluded from further analysis after conversion to nodes (hence the different values between 'raster' and 'node' demand/production). This aspect of the methodology will be elaborated upon in the updated version. Furthermore, Figure S2a, S2b and S2c should indeed reference figure S1(a,b, and c). This will of course be adapted in the next version.

7. The referee believes it will enhance the reliability of result if we compare the results with earlier estimates such as Bouwman et al. (2013) for manure; Van Drecht et al. (2009) and Moree et al. (2013) for domestic.

In the manuscript as is, we compare our results to three other related studies. We intend to further extend this comparison and create a more complete overview by including the works referenced to by the referee, also.

We would like to again thank the referee for his/her review and feedback. We hope to adequately address the issues identified in the updated version of the manuscript, and look forward to any other feedback he/she may have.

---

## Author Comment (AC2) · 30 Apr 2018

That authors would like to thank Dr. Yuan for his insightful messages on how to improve on the article. In the below, we would like to share our thoughts with regards to some of the presented concerns.

1. The referee remarks that the manuscript lacks an essential definition of what is understood as 'phosphorus wastes' in section 1.1.

In this study, we investigate the potential in partially meeting the agricultural phosphorus demand by recovering the phosphorus: i) discharged by humans through wastewater, and ii) excreted by animals through manure/liquid wastes. 'Phosphorus waste' by that definition is limited to only that which is produced by humans as accumulated in

wastewater streams, and that which is excreted by animals. The referee has made a very valid remark that this essential definition is missing. A more refined definition will be added in revised version of the article.

2. The referee claims to be confused about the methods used in determining global market price of phosphorus - as resulting largely from a lack of definition in the manuscript on exactly what product prices are evaluated for.

In the study, all prices (unless mentioned otherwise) are in U.S. dollars per tonne of elemental phosphorus: [\$ t-1] P. We tried to avoid presenting the prices in terms of [\$ t-1] phosphate rock, chemical fertilizer, struvite, or other forms of phosphorus (e.g. phosphate ($PO_4$)), so as not to confuse the reader and to make price estimates between the different products comparable. Even though, therefore, there are different products acting in the same phosphorus market, the model and actors judge the value of these products based only on the amount of absolute P in their contents, i.e. as [\$ t-1] P.

3. The referee remarks that 'phosphorus throughput rate' is not well defined with regards to formula (1) and that the recovery efficiency depends on the treatment technologies and collected rates (and other factors). The referee recommends to define and explain the production density definition and formula more clearly.

The authors recognize that, indeed, 'phosphorus throughput rate' is not well defined. In the study, 'phosphorus throughput rate' is presented as the kilograms of phosphorus per head, per year [kg h-1a-1] each organism of group O (human, cattle, poultry, swine), 'produce'. Since phosphorus is not 'produced' by organism, only consumed and excreted, the yearly amount of phosphorus excreted per head ([kg h-1a-1] P), is defined as the phosphorus throughput rate. This, or at least similar, definition will be included in the revised version of the article.

Furthermore, the recoverable amount indeed varies significantly with the composition of wastewater and manure. It is one of the major generalisations made in this study

that, for example, only 20% of phosphorus inflow into wastewater treatment facilities can be recovered via struvite precipitation. We will ensure that these and other related assumptions are presented in a comprehensive overview, e.g. in a table, and that these definitions are well explained in the revised version of the manuscript.

4. The referee remarks another issue with regards to a lack of definition for the 'phosphorus demand density'.

In the study, the phosphorus demand density is defined as the phosphorus which a crop patch requires in a year, and is expressed as tonnes per square kilometer [t km-2]. It further assumes no change in soil stored P. Therefore the total phosphorus demand per area is largely defined by the crop harvested area and how much phosphorus the crop requires per harvest as defined by FAO fertilizer manual. Furthermore, it does not include the phosphorus demand of any grazing animals or such. We will make sure to adequately define and describe these terms in the revised version of the article.

5. The referee remarks that in the section 2.2 phosphorus content is treated as only that which is found in wastewater and therefore does not include that found in solid wastes.

This study only focuses on the recovery of phosphorus from wastewater and therefore disregards any recovery potential of phosphorus from solid wastes. This is never explicitly defined as such, and therefore the confusion is understandable. In the revised version, we will be sure to define phosphorus waste in the introduction, as in accordance to point 1.

6. The author remarks to be totally confused about the trade model, and emphasizes that, to his best knowledge, phosphorus wastes are more suitably recycled locally and cannot be traded at global scale.

Firstly, we regret that we have not been able to explain the model adequately and will review our explanation of the model setup in the revised version of the manuscript.

Secondly, we believe our work shows that international trade in recovered phosphorus is feasible, just as international trade in chemical phosphatic fertilizers is feasible. In many ways, recovered phosphorus is similar to mined phosphorus. They both incur extraction costs, processing costs, and transportation costs. These costs determine the scale at which trade is feasible. Recovered products have a competitive advantage over mined products, as wastewater is almost everywhere where people are, while natural reserves are concentrated in few areas of the world (see Figure 1). Additionally, recovering phosphorus from wastewater can (depending on the wastewater treatment facility) lead to reduced operational costs (regardless of the sale of the product) thereby allowing these facilities to offer recovered products practically for free (as in The Netherlands). Indeed, however, because transportation cost plays an important role, our model also shows that most trade does occur locally/regionally, confirming the referee's intuition.

7. The author states that resolving the issues mentioned above will change the results entirely, requiring a new analysis and discussion of the implications.

We intend to review and revise the article in response to the comments of the referee. However, since most of the issues identified by Dr. Yuan pertain to (a lack of) definitions, we do not expect that it will change the results significantly.

8. The referee recommends to delete figure 1 (we assume, as the figure number is not specified) from the manuscript considering it presents basic data. Furthermore the referee regards Figure 2 to be confusing, wondering what is meant by phosphate rock; the production, the use or consumption, theoretic demand, or supply capacity? The referee further suggests to move the figure 3 and figure 4 to supporting information.

We thank the referee for the comment. Figures 1 and 2 were included in the introduction of the manuscript to contextualize the phosphorus market by highlighting and illustrating some of the important spatial (figure 1) and temporal (figure 2) aspects of the market. Figures 3 and 4 are integral model results for those interested in the market

potential of recovered phosphorus. We therefore intend to keep the figures as such.

9. The referee concludes that 'the english expressions need to be polished so as to make it more understandable with scientific basis'.

The authors recognize that the language can be adapted into a more scientific form and will polish the English in the revised version of the manuscript.

We would like to thank the referee for the feedback. We hope to adequately address the issues identified by Dr. Yuan to improve the manuscript and the study as a whole. We would be grateful to receive additional feedback, when and if it comes to mind.
* * *

---

## Referee Comment (RC3) · Anonymous Referee #3 · 16 May 2018

This manuscript analyses global potential of phosphorus recovery by developing a model which simulates phosphorus trade between supply and demand nodes. The topic is interesting for a wider audience and important in view of the fact that phosphorus is a non-renewable source and at the same crucial for agricultural production. The peak production of phosphate ore could occur already in the coming decades, which makes the assessment of recovery potential timely and relevant.

1. The authors could motivate better why there are using the minimum production costs for production nodes and maximum bid prices for demand nodes instead of average costs/optimal prices. I am wondering whether this potentially leads to an overestimation of trade because if average or maximum production costs were used, less trade would occur. The same is true if optimal or minimum bid prices of demand nodes are

used instead of maximum prices. Could this be the reason that the model results in a misbalance of 109% total production potential to demand?

2. The graphical representation of the supply and demand curves in Figure A in Supplementary Material does not seem correct. The quantity is typically shown on an x-axis and the price on the y-axis. Moreover, the demand and supply curves are usually concave. I would refer the authors to an economics textbook (e.g. by Samuelson and Nordhaus) for an appropriate description of the supply and demand functions.

3. For countries with high and intermediate urban access to sanitation, the assumed phosphorus recovery technology is different than for countries with low urban access to sanitation. It would be helpful if the authors reported the costs of these two different technologies. They could also comment on how the technological assumptions affect the results (e.g. how would the results change if the same technology is assumed in all countries?). Related to this, how realistic is it that only source separating-, dry composting toilets are implemented in countries with low urban access to sanitation? The authors could shortly elaborate on the social acceptance of such technologies and the potential yuck factor.

4. The model description is certainty important, but I found it difficult to follow. Is it possible to describe the model more concisely and clearly?

5. I have some doubts about how realistic the model is. What certainly gives it most credibility is the relatively good approximation of the market prices. However, the justification for selecting years 2005, 2006, 2011 and 2015 is missing. Is it possible to report the model performance for all years presented in Table 4?

6. The model includes the labour wage. How are these determined or which database is used for this purpose? Do labour wages vary across countries?

7. Please explain what the 'resource costs' in the model stand for.

8. This study does not seem to consider the external costs of phosphorus recovery,

such as CO2 emission and energy demand (see e.g. Linderholm et al., 2012). It is recommendable to explicitly mention this and briefly discuss the implications on results. Of particular interest would be to know which of the two technologies considered in this study has higher external costs.

9. The environmentally extended world input-output tables unfortunately do not include data on phosphorus yet. Nevertheless, I was wondering if the elements of an input-output analysis or computable general equilibrium models could contribute to a more precise and realistic representation of the phosphorus supply, demand and in particular trade? Would the authors consider this as an interesting area for future research or do they not see the added value of it?

10. Two important aspects are missing in my view when explaining the far-future scenario: the prices of phosphorus are expected to be high and the new technologies (in addition to the existing ones) are likely to become available and economically feasible.

11. The literature overview with respect to phosphorus recovery could be more complete (e.g. Cornel and Schaum, 2009; Cordell et al., 2011; Molinos-Senante et al., 2011).

12. It would be useful to mention early on that the economic feasibility of P recovery varies in space because it depends on the concentration of P in wastewater, which is related to the population and livestock density. This connection was not clear to me immediately.

13. Throughout the paper I was getting confused with the term 'production', as in phosphorus production potential. Since phosphorus is produced in rock formations, do authors actually mean 'recovery' potential? I think it is important to distinguish between the terms 'production' and 'recovery' to avoid confusion.

14. Proofreading the manuscript for English language is highly recommendable.

References: Cordell, D., Rosemarin, A., Schröder, J.J., Smit, A.L. (2011). Towards
global phosphorus security: A systems framework for phosphorus recovery and reuse options. Chemosphere 84: 747-758. Cornel, P., Schaum, C. (2009). Phosphorus recovery from wastewater: needs, technologies and costs. Water Science and Technology 59: 1069-1076. Linderholm, K., Tillman, A.-M., Mattsson, J. E. (2012). Life cycle assessment of phosphorus alternatives for Swedish agriculture. Resources, Conservation and Recycling 66:27-39. Molinos-Senante, M., Hernandez-Sancho, F., Sala-Garrido, R., Garrido-Baserba, M. (2011). Economic feasibility study of phosphorus recovery processes. Ambio 40: 408–416.

---

## Referee Comment (RC4) · K. Udert (Referee) · 24 May 2018

**GENERAL COMMENTS**

By using statistical data for populations and agriculture and assuming basic equations for phosphorus recovery and agricultural phosphorus demand, the authors can give reasonable estimates for the potential of the market potential for recycled phosphorus. The results compare very well with empirically determined costs and prices.

The results of this study add important information for the discussion on phosphorus recycling and they also show issues, which have to be addressed in phosphorus recycling. For example, they show that transport costs are crucial and therefore high concentrated recycling products are needed or short distances. I recommend to accept this manuscript for publications after the manuscript will be revised according to the comments below.

SPECIFIC COMMENTS

Sensitivity of input data: Some information about the sensitivity of the input data would help to support the discussion. Especially the influence of the P recovery efficiency could have a major effect on the result. With struvite, the authors chose a P recovery process, with a very low recovery efficiency. See e.g. Egle, L., Rechberger, H., Krampe, J. and Zessner, M. (2016) Phosphorus recovery from municipal wastewater: An integrated comparative technological, environmental and economic assessment of P recovery technologies. Science of the Total Environment 571, 522-542

Global P demand and global P recovery: It is surprising but also very interesting that the global P demand fits the global P recovery so well. It would be good if the authors could compare their data with a phosphorus balance as provided by Cordell et al. (2009), 8Scholz et al. (Scholz, R.W., Roy, A.H., Brand, F.S., Hellums, D.T. and Ulrich, A.E. (2014) Sustainable Phosphorus Management. A Global Transdisciplinary Roadmap, Springer, Dordrecht.) or van Dijk et al. (van Dijk, K.C., Lesschen, J.P. and Oenema, O. (2016) Phosphorus flows and balances of the European Union Member States. Science of the Total Environment 542, 1078-1093.). With such a comparison, the reasons for possible deviations could be identified. This discussion would be extremely helpful. One reason, which could explain the deviation could be the use of P accumulated in the soil for plant growth.

Free market: This study considers a free market for agricultural products, but in reality, this market strongly influenced by subsidies. How could this be considered?

Free struvite from Bio-P and setting of the costs for wastewater treatment: I am missing a comparison between the costs of struvite production in Bio-P wastewater treatment plants and the costs caused by struvite blockages. The authors assume that struvite recovery from wastewater is free because it off-sets the costs for removing blockages,

but is this correct? Along these lines, it might be good to mention that P recovery from dry sanitation systems would also help to reduce environmental pollution. This is a value for society and would reduce the costs of the P fertilizer production. However, I understand that assessing this value is not easy.

Conclusions: The "Conclusions" section is rather a summary. Conclusions do not necessarily repeat the main results from study but they should rather emphasize the significance of the results.

Formulae: The formulae should be revised. I added a few remarks in the detailed comments. A formula on the production costs for compost is missing and should be added.

Language: The text contains several errors. I noted some of them in the detailed comments below. The introduction should be written more clearly. Some more hints are given in the detailed comments.

DETAILED COMMENTS

Page 1 Line 8 Sentence is unclear

Page 1 Line 9 "as well as"

Page 1 Line 11 It is unclear what the subclause "while rock phosphate products exist" means.

Page 1 Line 18 Write "Ammonium" and "Phosphate" in small letters.

Page 1 Line 22 This sentence is unclear. Could you give a reference? Much of the phosphorus used today is flushed to the sea and forms sediments. This is the first step to form sedimentary phosphate rock.

Page 1 Line 24 The estimation that peak phosphorus might occur in 2030 is based on several wrong assumptions. It is unlikely that a peak phosphorus will occur in the near future. The reserves are much larger than it was known to Cordell et al. and there are

large resources, which still need to be explored. See Scholz, R.W. and Wellmer, F.-W. (2013) Approaching a dynamic view on the availability of mineral resources: What we may learn from the case of phosphorus? Global Environmental Change 23(1), 11-27.

Page 1 Line 26 Depletion of phosphorus is presumably not the issue but rather economic scarcity, which is mainly due to developments on the financial market. However, these developments affect not only phosphorus but also other commodities. See the reference give above.

Page 1 Line 27 "accessibility"

Notes on page 1 I suggest to include these notes in the text.

Page 2 Line 6 Delete "disproportionately", because the meaning is unclear. What would be a "proportionately large algal bloom"?

Page 2 Line 7 Replace "water based" with "aquatic".

Page 2 Line 25 "Favourable economic prospectives..." This sentence is unclear to me. You probably mean "perspectives".

Page 2 Line 30 The whole paragraph is hard to understand and should be reformulated.

Page 3 Line 10 Phosphorus-laden wastewaters and agricultural areas do not "concentrate themselves". There are locations with high concentrations are they are concentrated at certain locations.

Page 3 Line 13 What do you mean by "paying prices of varying crop sites"? The money that farmers will get for their crops?

Page 3 Line 22 Reverse the sentence: A crude, global mapping of phosphorus production sites is achieved by using..."

Page 3 Line 25 Do you mean "flow" when you write "throughput"?

Page 3 Line 23 "formula"

Page 3 line 31 Rather use "cap" for "capita" instead of "h" for "head". "h" is usually used for "hours".

Page 3 Line 36 "phosphorus"

Page 4 Line 6 Check superscripts

Page 4 It would be helpful to explain the parameters used in equation 2 and 3 in more detail, especially, how the values for these parameters are derived.

Page 4 Line 18 Explain better, what you mean by pixel.

Page 4 line 26 "marked" instead of "stylized"

Page 4 Line 35 "sanitation"

Page 5 Line 3 The struvite process has probably the lowest P recovery efficiency of all currently known P recovery processes. See e.g. Egle, L., Rechberger, H., Krampe, J. and Zessner, M. (2016) Phosphorus recovery from municipal wastewater: An integrated comparative technological, environmental and economic assessment of P recovery technologies. Science of the Total Environment 571, 522-542. Ideally, the authors would also include another process with higher phosphorus recovery to show the potential of P recovery. Alternatively, a sensitivity analysis on the P recovery efficiency could be included.

Page 5 Line 13 Is there also struvite precipitation on the livestock farms?

Page 5 Line 17 90% P recovery from urine can only be achieved, if the P, which precipitates in the pipes and collection tanks is also recovered. This spontaneously precipitated P can make up 30% of the total P.

Page 5 Line 25 What do you mean by resource cost?

Page 5 Line 30 Give some more information about the literature reference of Egle. At which university was the PhD thesis conducted and when was it concluded?

Page 5 Line 30 Why did you choose MgCl2 and not MgO as magnesium source? MgCl2 is more expensive. It would be good to include a sensitivity analysis on the Mg source price.

Page 5 Line 33 Does the production cost cover exactly the cost for struvite and sludge handling?

Page 5 Line 34 How do you include the additional resource cost? Is it part of BS?

Page 5 Line 35 "as follows"

Page 5 Line 4 E_Lt must be larger. This value corresponds to 1.6 L/100 km, which is extremely low. I would guess that it might be at least a factor 10 larger. W_Lt seems to be high. This might be the combined weight for truck and load.

Page 6 Equation 6 Doesn't equation 6 also have to contain transportation costs (see the term in equation 7)?

Page 6 Equation 7 Please check this equation. The units do not fit. The term Pb*EW results in [$/d], while the term CF/VW has the unit [$/km]. What is a "bunker" and what is a "handy size bulk carrier"?

Page 8 Equation 11 These equations should be adapted because it is not mathematically correct. DQn and SQi cannot stand on both sides of the equation.

Page 8 Equation 12 Does this equation mean that the population grew at the same rate at every node? What does "population density raster" mean?

Page 10 Line 1 "determine how"

Page 10 Line 7 The interpretation of the box plots is already described in the caption text. This description does not have to be repeated in the main text.

Page 10 Line 31 Delete on "pay"

Page 11 Line 8 "treatment"

Page 11 Line 9 "variable" instead of "vary"

Page 11 Line 10 "feasibility"

Page 11 Line 16 Give the number always in the same way, e.g. in brackets.

Page 11 Line 30 "distinctly"

Page 11 Line 38 Why are the prices given in Figure 4 so much higher?

Page 11 Line 39 "Phosphorus"

Page 12 Line 30 "phosphorus"
* * *

---

## Referee Comment (RC5) · Anonymous Referee #3 · 25 May 2018

6. A wage of 17$/h seems a lot for a global average. Could you include a reference that supports this figure?

7. The production costs usually consist of fixed and variable costs. So, in this study the resource costs fall into the 'variable costs' category. The terms fixed and variable costs would be more appropriate. Explaining what resource costs entail is still recommendable.

---

## Short Comment (SC1) · 25 May 2018

6. We agree that a globally homogenous labor wage of 17 [$/h] for a truck driver is very likely a too conservative estimate. We will run the model at lower wages (10-15) and see how this impacts the results.

The original 17 [$/h] value is an informed estimate we made provided the absence of a global dataset on such labor wage per country or even per region. We had instead taken into consideration the wage indicators from the U.S. ($\sim$20 $/h) (1), the Netherlands ($\sim$30$/h) (2), and China ($\sim$6 $/h) (3) (mostly from non-official sources).

Hopefully the sensitivity analysis will provide meaningful insight on how the uncertainty around the chosen wage will affect the results.

[Figure]

7. We agree that defining our cost equations in terms of fixed and variable costs is an improvement to the comprehensibility of the price equations and therefore the study as a whole. We will be sure to adapt the text and formulas accordingly and to define more precisely also what is implied by resource costs (e.g. the $MgCl_2$ required for struvite precipitation).

Thank you again for elaborating on these issues. We want to make sure the final manuscript is as clear as possible, especially provided the extent and complexity of the methodology.

(1): https://www.bls.gov/oes/current/oes533032.htm;

(2): https://www.mijnzzp.nl/Beroep/608-Vrachtwagenchauffeur/Salaris-en-tarief;

(3): https://www.salaryexpert.com/salary/job/heavy-truck-driver/china;

---

## Author Comment (AC3) · 25 May 2018

The authors would like to thank the anonymous referee for the constructive feedback and welcomed contributions. In the below, we would like to present our a summary of our response and the motivations for some of the decisions made.

1. The referee suggests presenting a motivation as to why we chose to use minimum production costs for production nodes and maximum bid prices for demand nodes instead of average costs/optimal prices. The referee wonders whether this potentially leads to an overestimation of trade, resulting in the misbalance of 109% total production to demand.

The rationale for this based on the assumption that both the producers and consumers

maximize their profits. A producer will then be willing to recycle P only if it is able to atleast recover its cost of producing it. Meanwhile, a consumer will only be able to buy as long as its profit is above zero. Thus the maximum price of recycled P that it is willing to pay is the marginal value of the crop that it grows, i.e. so-called maximum bid price. The space between these two prices (as long as minimum production cost is lower than the maximum bid price) is the space to negotiate and settle the price at which recycled P is traded.

The 109% imbalance the referee mentions derives from a balance made purely using the statistical throughput figures and population density maps for production, and crop phosphorus requirements and crop harvest area maps for demand. It is a summation of the estimated total livestock (cow, chicken, pig) phosphorus excretion and human phosphorus excretion rates, divided by the sum phosphorus demand of all agricultural areas. Since these production/throughput rates are taken as average (and in reality vary with age, diet, gender, and species for livestock and crops, etc), some error in this balance was imminent. Also, there are some major phosphorus consumers not included, such as the pharmaceutical industry and detergent manufacturers. In reality, these also have an at least 10% share of the actual global phosphorus demand. Including this 10% would likely balance the budget almost perfectly despite the generalizations. However, since we do not account for these actors in this study, we decided to just take the imbalance as is. We will definitely explain this imbalance in more detail in the revised draft to avoid future confusion.

2. The referee remarks the supply and demand curves in Figure A of the Supplementary Materials have inverted axis and ought to be concave instead of convex.

We agree that the figure is not entirely correct, the demand curve is convex while it is expected to be concave (however supply curve is expected to be convex as shown). We will adapt the illustrative figure accordingly. However, we intend to keep the axes as such since it does not influence the economic interpretation.

3. The referee recommends to elaborate on the costs and implications of applying different technologies for different development regions. The authors agree that this is indeed insufficiently discussed in the manuscript. Although much of this discussion was intentionally left out to shorten the length of the manuscript, we agree that it is an important aspect of our study and therefore needs to be elaborated upon in the next revision of the article.

4. The referee suggests to rework the model description which is currently difficult to follow. The authors recognize that the model description can be improved and will attempt to do so in the revised version.

5. The referee has doubts about how realistic the model is, and cannot find justification for selecting the simulation years 2005, 2006, 2011 and 2015. The referee recommends running the model for all years.

The authors acknowledge that the model performance can be validated more rigorously. We will present model estimates for all the years. Also, we will include a model sensitivity analysis (see referee #1 comment) in the next draft.

6. The referee asks how labour wage is determined.

In the study, we assume a globally homogenous labour wage of 17 [\$ h-1] for truck drivers and cargo ship personnel. Although trucking wage in the U.S. is estimated at around 20 [\$ h-1], we arbitrarily assumed a slightly lower value considering that wages in the vast majority of the world are likely lower. We will explain this in the revised draft.

7. The referee asks to explain what 'resource cost' stands for.

The production cost of a production node has two main components: 1) The technology investment cost (fixed value for each technology) and, 2) a resource cost. While the investment cost is fixed per x number of people (i.e. 1 mio \$ per 500,000 people), the resource cost varies with the amount of product recovered (i.e. the cost of magnesium chloride for struvite precipitation ~740 [\$ t-1] P recoverable). We will explain in greater

detail what the 'resource cost' entails in the revised version of the manuscript.

8. The referee notices that the study does not consider external phosphorus costs such as CO2 emissions and energy demands, and recommends to explicitly mention these.

The authors acknowledge that these costs are relevant for consideration and discussion. Some assessment with regards to the difference in total CO2 emissions was included in earlier versions of the model, but was later abandoned to reduce the (still very large) scope of the model. The authors will briefly discuss these external costs in the revised version.

9. The referee wonders if environmentally extended world input-output (EEIO) studies on phosphorus could make a valuable contribution to a more precise and realistic representation of phosphorus supply, demand, and in particular trade.

The authors believe that all studies on phosphorus, of whatever nature, can at this time make extremely valuable contributions to the effort of stimulating the transitioning to a sustainable phosphorus market in general, and that EEIO studies can play an especially important role in this at a national spatial resolution. However our model runs at finer resolution, so EEIO based datasets can be used to constrain sub-national (grid) scale estimates of P supply and demand. We will discuss this possibility in our revised version.

10. The referee mentions that two important aspects for the far-future scenario that are missing in his view: 1) the high prices of phosphorus, and 2) the availability of new and cheaper technologies.

The authors acknowledge that discussion of these dimensions is not present currently. Although the prices of phosphorus are determined to rise in our future scenario (in accordance to the referee's intuition and other study predictions), we do not account for a decrease in recovery costs due to technological innovation, or the introduction

of new technologies to the market. Since the degree of innovation and extent of new technological developments are uncertain, we ignored this aspect all together in order to not complicate the study further. The authors definitely agree, however, that this merits much more attention and therefore this concept will be discussed in the revised version.

11. The referee references three additional sources that should be cited with regards to phosphorus recovery. The authors will consider these publications and cite them in the manuscript where relevant.

12. The referee recommends to introduce early on that the economic feasibility of P recovery varies in space as it depends on the concentration of P in wastewater, which is related to the population and livestock density.

The authors would like refer the referee to section 1.1, line 19 of page 2, where this concept of spatial variability is introduced quite early on. To bring this to greater attention, however, we will see if we can draw the link to population density also at this point, without adding to many additional words or unnecessarily overcomplicating the introduction.

13. Similar to referee #2, the referee (#3) is led into confusion by improper definition and inconsistent use of terminology. The authors recognize this issue and will address this in the revised version.

The authors would like to sincerely thank the anonymous referee for his well-grounded feedback. We hope we have understood correctly- and responded adequately to the referees questions, and we look forward to improving the manuscript accordingly.

---

## Author Comment (AC4) · 25 May 2018

The authors would like to thank Prof. Dr. Udert for the constructive feedback provided in his concise review. In the text below, we would like to present our proposed strategy to adapt the manuscript with regards to the recommendations made and issues identified.

1. The referee notes the absence of sensitivity analysis. A sensitivity analysis is believed to be important especially considering the low efficiency of the phosphorus recovery technology analyzed in this study.

The authors are currently conducting a sensitivity analysis to include in the revised draft. In accordance the referee's recommendation, we will evaluate the model sensitivity to the recovery efficiency and discuss this with respect to the recovery efficiencies

of other recovery technologies based on data provided by Egle et al. (2016).

2. The referee recommends comparison of phosphorus recovery potential and demand balances with those of other studies.

To keep the article concise, and provided the many other products that merit in-depth evaluation and comparison, little attention was paid to evaluating the phosphorus recover potential and demand balances in the original manuscript. However, recognizing that it does form one of the integral components of the methodology that determines recovery feasibility and trade, we acknowledge that some comparison with other studies should be made. We will do so in the next revision of the manuscript.

3. The referee questions what role agricultural subsidies could play in influencing the crop market, and therefor phosphorus market.

Agricultural subsidies may be important in influencing crop market dynamics. If crop subsidy data at national or regional level is available (in reducing production cost ([$/t]) per crop type) then this could be quite simply incorporated into the model. A subsidy would result in an increase in a node's maximum bid price, making it more competitive in the market. This improvement would allow the model to not only differentiate 'purchasing power' based on low and high value crops, but also (realistically) between different agricultural regions and subsidy policies. This differentiation would be especially interesting for the far-future scenario, when phosphorus supply is limiting and agricultural demand is high. The current maximum bid price equation, however, is already very simple and does not account for many other regional differences (cost of equipment, labor wage, farmer living standards, work efficiency, subsidies etc.). Therefore the entire demand side requires reevaluation - something that is planned for future version of the model, but will likely not be feasible to include already in the next version of the manuscript.

4. The referee misses a comparison between costs and cost savings from struvite crystallization, where cost savings result from reduced struvite scaling maintenance

work. Also, the referee recommends to mention that P recovery from dry sanitation systems would also help to reduce environmental pollution, which is a value for society and would reduce the costs of the P fertilizer production.

The authors will present a brief overview of the costs and savings of struvite precipitation based on other literature in the revised version of the manuscript. With regards to the external costs and benefits of recovery, we would prefer to keep the manuscript purely monetary in terms of its economics but will be sure to emphasize the additional value environmental benefits (e.g. in terms of water use and pollution) that some recovery options may offer.

5. The referee remarks that the conclusion is more a summary of results than a conclusion emphasizing the significance of these results.

The authors agree that the current conclusion may be more appropriately titled 'summary', and will therefore change this in the revised draft. We will also draft a separate conclusion that emphasizes the significance of the results and highlights the issues that have to be addressed in phosphorus recycling in general.

6. The author lists a series of recommended changes with regards to language and formulae.

We are grateful for detailed feedback and will include all recommendations in our revision.

Furthermore, we would like to sincerely thank Prof. Dr. Udert for taking the time to provide a meaningful contribution to improving the manuscript. We will adapt the manuscript accordingly and look forward to any other comments should they come to mind.

Egle, L., Rechberger, H., Krampe, J. and Zessner, M. (2016) Phosphorus recovery from municipal wastewater: An integrated comparative technological, environmental and economic assessment of P recovery technologies. Science of the Total Environment

571, 522-542

---

## Referee Comment (RC6) · Z. Yuan (Referee) · 26 May 2018

I suggest the revised manuscript can be uploaded and reviewed again, rather than just discussions because some questions can not be answered with scientific methods even if they can be discussed again and again? With regard to the authors' response, I just want to further comment as only one thing first:

The authors think that the issues I concerned belong to definitions and they do not think that it will change the results significantly. here I just give one example to prove that it is so important to define the scope and it definitely will change the results. The author response that "phosphorus wastes" means 1)discharged by humans through wastewater, and 2)excreted by animals through manure/liquid wastes. and further explained

that it is limited to that produced by humans as accumulated in wastewater streams, and that excreted by animals. I would like to confirm that if the food wastes, kitchen garbage, sludge of wastewater plants, residues of animal slaughtering and feed-stuff processing were considered in this manuscript? if these wastes were included in your definition? If you can define the phosphorus wastes clearly and revise the manuscript according to your definition, I would like to confirm if it change your results and I would really like to review it again based on your explanations.

---

## Author Comment (AC5) · 11 Jun 2018

Thank you Dr. Yuan for your review. In order to avoid confusing potential/future readers, we will do our best to clarify the introduction and methodology, as well as specifiy the definition of all general and specific terms related to the study (likely in-text, but possibly in an appendix).

---

## Author Comment (AC6) · 11 Jun 2018

My apologies to the anonymous referee, as my previous response was incorrectly posted as a 'short comment' - therefore not counting as an official rebuttal argument in the discussion thread and inhibiting closure of the interactive discussion. Please find a copy of the response below, should you have missed the initial 'short comment'.

""" 6. We agree that a globally homogenous labor wage of 17 [$/h] for a truck driver is very likely a too conservative estimate. We will run the model at lower wages (10-15) and see how this impacts the results. The original 17 [$/h] value is an informed estimate we made provided the absence of a global dataset on such labor wage per country or even per region. We had instead taken into consideration the wage indicators from the

[Figure]

U.S. (c.a 20 \$/h) (1), the Netherlands (c.a 30\$/h) (2), and China (c.a 6 \$/h) (3) (mostly from non-official sources). Hopefully the sensitivity analysis will provide meaningful insight on how the uncertainty around the chosen wage will affect the results.

7. We agree that defining our cost equations in terms of fixed and variable costs is an improvement to the comprehensibility of the price equations and therefore the study as a whole. We will be sure to adapt the text and formulas accordingly and to define more precisely also what is implied by resource costs (e.g. the MgCl2 required for struvite precipitation). Thank you again for elaborating on these issues. We want to make sure the final manuscript is as clear as possible, especially provided the extent and complexity of the methodology.

(1): https://www.bls.gov/oes/current/oes533032.htm;

(2): https://www.mijnzzp.nl/Beroep/608-Vrachtwagenchauffeur/Salaris-en-tarief;

(3): https://www.salaryexpert.com/salary/job/heavy-truck-driver/china;

""" -Dirk-Jan Kok, 25 May 2018

We are again grateful for the referee's time and valid feedback, and we appreciate the effort made in helping to improve the manuscript.

---

## Author Response (AR1)

Dear Editor,

This document provides a point-by-point summary of the feedback received from all referees with regards to the publicly reviewed manuscript 'Global Phosphorus Recovery for Agricultural Reuse'. The first section presents the general and frequently recurrent feedback. This is then followed by a chronological addressing of each of the referee's specific feedback.

Please also find attached with this document: 1) the revised manuscript, and 2) a marked-up document showing all changes made.

Sincerely,

On behalf of the authors,
Dirk-Jan Kok
* * *
**General Feedback** recurrent in the reviews of at least two of the referee's:

1. Absence of an uncertainty analysis with regards to modeling.

   Section 3.3.3 is adapted and now titled 'Model Validation and Sensitivity Assessment'. It covers briefly the results of a model sensitivity analysis and refers the reader to the supplementary materials for data on individual parameter sensitivities.

2. Language and spelling

   The text has been spelling-checked and proof-read an additional time by the author and coauthors.

3. Difficult to follow model description

   We have adapted our explanation of the trade model in section 2.3.2., 'Quantification of trade flows' (page 10), so that the model may be better understood. Furthermore, one of our main findings confirms that most phosphorus is recycled locally within the province/region. This is now highlighted in line 4 of page 16, in the conclusion.

**Feedback Referee 1**

1.  The referee identified a lack in detailed explanation on how we determine the P coming from manure.

    Line 28 of page 4 now explains that we follow the methodology provided by Sheldrick et al., (2003), and gives some background info as to what this method entails.

2.  The referee found equation 2 confusing and remarks that no detail on the input is provided. The referee also questions the introduction of the additional parameters (Tg, AH, and C) into the equation.

    Paragraph starting on line 13 on page 5 now refers the reader to a table in the supplementary materials that provides details on the input. Paragraph starting on line 21 now explains why it was necessary to include the additional parameters into the equation.

3.  The referee found it unclear from the manuscript how the demand and supply nodes are defined, and more specifically how we determine their location.

    Section 2.1.3 'From raster to nodes' is introduced (page 6) which explains how we define and determine the location of the nodes.

4.  The referee wondered if we account for the legacy P or the residual P that accumulates in the soil.

    Line 31 on page 5 now explicitly states that we assume no change in soil stored phosphorus.

5.  See point 1 of 'General Feedback'

6.  The referee remarked that Table 1 is not clear, and wonders why the values for Livestock Raster vs Livestock node and Human raster vs urban node are different? He also remarks that he is unable to find Figure S2a, S2b, and S2c.

    We have removed the incorrect reference to figure S2a, S2b and S2c from the text. We have adapted the contents and caption of table 1 to define more clearly what is presented.

7.  The referee believes it will enhance the reliability of result if we compare the results with earlier estimates such as Bouwman et al. (2013) for manure; Van Drecht et al. (2009) and Moree et al. (2013) for domestic.

    We expand our comparison of results to include to authors mentioned by the referee (paragraph starting at line 24 on page 14).

**Feedback Referee 2**

1. The referee remarked that the manuscript lacks an essential definition of what is understood as 'phosphorus wastes' in section 1.1.

   The emphasis on wastewater is now stressed throughout section 1.1. Furthermore, paragraph 2 of section 1.2 now explicitly states that the project focuses only on wastewater and only through recovery by struvite precipitation means.

2. The referee was confused about the methods used in determining global market price of phosphorus as resulting largely from a lack of definition in the manuscript on exactly what product prices are evaluated for.

   At the beginning of section 2.2, 'Determination of Node Prices' (page 6) we now explicitly define what we determine price for (line 20).

3. The referee remarks that 'phosphorus throughput rate' is not well defined with regards to formula (1) and that the recovery efficiency depends on the treatment technologies and collected rates (and other factors). The referee recommends to define and explain the production density definition and formula more clearly.

   Line 14, page 4, now explicitly defines phosphorus throughput rate. Line 31 explains that the recovery efficiencies depend on the technology, and that we assume recovery rates reported by (Sheldrick et al., 2003). We adapted the paragraph both proceeding and following the production density formula so that it explains better what it means.

4. The referee remarks another definition issue with regards to the 'phosphorus demand density'

   Phosphorus demand density is now defined in line 1 of page 5.

5. The referee remarks that in the section 2.2, phosphorus content is treated as only that which is found in wastewater and therefore does not include that found in solid wastes.

   Line 6 of page 3 now explicitly states that we do not consider solid wastes in this investigation.

6. See point 3 of the 'General Feedback'

7. The author states that resolving the issues mentioned above will change the results entirely, requiring a new analysis and discussion of the implications.

   We have adapted our definitions according to the original project scope. Results and analysis have been adjust following the feedback of all referees.

8. The referee recommends to delete figure 1 (we assume, as the figure number is not specified) from the manuscript considering it presents basic data. Furthermore the referee regards Figure 2 to be confusing, wondering what is meant by phosphate rock; the production, the use or

consumption, theoretic demand, or supply capacity? The referee further suggests to move the figure 3 and figure 4 to supporting information.

After discussion with the co-authors, we have decided to keep the figures the referee recommends to remove. Although basic data, they stress the most important premise of this research which is that the feasibility of phosphorus recovery is spatially and temporally dependent and therefore has potential. Provided that figures 3 and 4 form an integral part of the results, we also decided to keep these.

**Feedback Referee 3**

1. The referee suggests presenting a motivation as to why we chose to use minimum production costs for production nodes and maximum bid prices for demand nodes instead of average costs/optimal prices. The referee wonders whether this potentially leads to an overestimation of trade, resulting in the misbalance of 109% total production to demand.

   Line 27 and 28 of page 12 now stresses that prices are not included in the analysis before point. The previous results (such as the balance), therefore do not consider prices. This is now in line 31 on page 11: " indication of the *total potential* of recovering all excreted phosphorus" (i.e. not that of trade, and therefore not including prices).

   The 109% imbalance the referee mentions derives from a balance made purely using the statistical throughput figures and population density maps for production, and crop phosphorus requirements and crop harvest area maps for demand. This is determined before considering prices and economics.

2. The referee remarks the supply and demand curves in Figure A of the Supplementary Materials have inverted axis and ought to be concave instead of convex

   The axis are inverted and trends have been made concave instead convex.

3. The referee recommends to elaborate on the costs and implications of applying different technologies for different development regions.

   We dedicate most of page 7 to elaborating on the technology choices and costs for each development region. We also discuss the implications in table 3, 'Overview of assumptions'.

4. See point 2 'general feedback'

5. The referee has doubts about how realistic the model is, and cannot find justification for selecting the simulation years 2005, 2006, 2011 and 2015. The referee recommends running the model for all years.

   We have included a sensitivity analysis (section 3.3.3, page 13) and have now run the simulation for all years (2005 to 2015; see Table 2.)

6. The referee asks how labor wage is determined.

   We explain to the referee that we cannot pinpoint a globally homogenous labour wage, but that we estimate the one we've used based on non-official source on the internet. A summary of input data and the sources or inspirations is listed in the Supplementary Materials.

7. The referee asks to explain what 'resource cost' stands for.

By the recommendation of the referee (later response), we have renamed the term 'variable costs'.

8. The referee notices that the study does not consider external phosphorus costs such as CO2 emissions and energy demands and recommends to explicitly mention these.

    In line 10 of page 14, we present an explanation as to why we didn't include external costs and provide a brief assessment of the implications thereof.

9. The referee wonders if environmentally extended world input-output (EEIO) studies on phosphorus could make a valuable contribution to a more precise and realistic representation of phosphorus supply, demand, and in particular trade.

    We believe that all studies on phosphorus, of whatever nature, can make extremely valuable contributions to the effort of stimulating the transitioning to a sustainable phosphorus market in general, and that EEIO studies can play an especially important role in this at a national spatial resolution. However our model runs at finer resolution (sub-national), and therefore EEIO datasets are not easily integrated into- or compared with the results of this study.

10. The referee mentions that two important aspects for the far-future scenario that are missing in his view: 1) the high prices of phosphorus, and 2) the availability of new and cheaper technologies.

    We now emphasize in our conclusion (line 10, page 16) that future scenarios will change as existing and new technologies develop in the future. Also, the table of assumptions now mentions how it could impact the results if we accounted for this change.

11. The referee references three additional sources that should be cited with regards to phosphorus recovery.

    We have reference the recommended authors.

12. The referee recommends to introduce early on that the economic feasibility of P recovery varies in space as it depends on the concentration of P in wastewater, which is related to the population and livestock density.

    Line 25 to 29 (page 2) now emphasize the relevance of the spatial scale to assessing economic feasibility. Line 31 now explicitly states that we intend to "evaluate the *spatially dependent* feasibility for phosphorus recovery from wastewater". Line 18, page 4, introduces the importance of population density maps.

13. The referee (#3) is led into confusion by improper definition and inconsistent use of terminology.

    The authors recognize this issue and have defined the terms more explicitly. See also feedback referee 2, point 5 and 6.

**Feedback Referee 4**

1. See point 1 of the 'General Feedback'

2. The referee recommends comparison of phosphorus recovery potential and demand balances with those of other studies.

   We have expanded our comparison of results to seven other studies (Paragraph starting at line 24, page 14)

3. The referee questions what role agricultural subsidies could play in influencing the crop market, and, therefore, phosphorus market.

   We mention that subsidies could affect the maximum bid prices of agricultural nodes in the assumptions listed in Table 2.

4. The referee misses a comparison between costs and cost savings from struvite crystallization, where cost savings result from reduced struvite scaling maintenance. Also, the referee recommends to mention that P recovery from dry sanitation systems would also help to reduce environmental pollution, which is a value for society and would reduce the costs of the P fertilizer production.

   The resulting difference in struvite price by either including or excluding scaling maintenance savings is observed in figure 3 where the reader can compare the price for Urb 1-Str. (incl. savings) vs. Urb 2-Str (excl. savings). Furthermore, making a pure monetary assessment, we do not include external costs or benefits such as those to the environment (line 10 of page 14). We do end the manuscript emphasizing the importance of achieving "maximum benefits […] for both the environment and the urban community, as well as the livestock and agricultural sectors".

5. The referee remarks that the conclusion is more a summary of results than a conclusion emphasizing the significance of these results.

   We have removed the summary of the results from the conclusion. We have adapted the conclusion to now only emphasize the significance of the results as presented in the results section.

6. The author lists a series of recommended changes with regards to language and formulae.

   We have adopted all the recommended changes to language and formulae.

Summary
24/07/2018 23:21:21

Differences exist between documents.

**New Document:**
Phosphorus_Recovery_Article_v3
28 pages (853 KB)
24/07/2018 23:21:01
Used to display results.

**Old Document:**
Phosphorus_Recovery_Article_v2
18 pages (696 KB)
24/07/2018 23:21:01

No pages were deleted

**How to read this report**

**Highlight** indicates a change.
 indicates deleted content.
▲ indicates pages were changed.
⬌ indicates pages were moved.

[revised manuscript text omitted]

---

## Author Response (AR3)

Dear Editor,

This document provides a point-by-point summary of the feedback on the manuscript 'Global Phosphorus Recovery for Agricultural Reuse'

Please also find attached with this document: 1) the revised manuscript, and 2) a marked-up document showing all changes made.

Sincerely,

On behalf of all authors,
Dirk-Jan Kok
* * *
**Editor Feedback**:

1. Mathematical notations: Please adhere to the HESS requirements. Remove for example the "*" symbol for multiplication in all equations and brackets around units if they are a normal part of the text.

    We have replace asterix multiplication symbol with the recommended '·' symbol.
    We have ensured that all units in text are in brackets

2. p. 1: Move "through struvite crystallization" (p. 1, L. 14) to p. 1, L. 8 (after "wastewater").

    Moved "through struvite crystallization" from p. 1, L. 14 to p. 1, L. 9.

3. p. 4, L. 4: Remove comma after "objective"

    Removed comma after "objective"

4. p. 4, L. 11: "organisms" instead of "organism"

    Corrected spelling of "organisms"

5. p. 4, L. 13 (and elsewhere): What does the symbol "h" stand for in the bracket?

    Replaced symbol 'h' with 'head' (per individual).

6. p. 4, L. 15: The expression "person individual" sounds weird.

    Corrected "person individual" to '[…] each individual person'

7. p. 13, L. 23 – 25: "relative accuracy": Please provide a quantitative description of how well the model performs.

> Removed "relative accuracy". The paragraph now presents the numerical differences between predicted and observed prices.

8. p. 13, L. 33: Table S7 is actually a figure.

> Moved to figures section and renamed in text.

9. P. 13, L. 33 – 37: This paragraph is very superficial. Screening through the results that you show in the SI, I think some more findings should be reported. As it appears now it is more an exercise to demonstrate that a sensitivity analysis has been carried out but you do not really provide much insight for the reader. However, your figures reveal quite diverse patterns ranging from a complete absence of sensitivity for certain variable pairs, over linear relationships to very non-linear patterns. Please provide some essential findings and refer for explicitly to specific data that you show in the Supplementary Material.

> We now present the figure data also in a table (Figure S3 → Table S7).
> We elaborate on the implications of the sensitivity analysis in the Sup. Mats. and mention some essential findings in order to provide insight for the reader.

10. Fig. 3: Please increase the font size for better readability.'

> We increased the fontsize.

11. Fig. 4: Please change the colour set you use. It is hardly possible to distinguish the different categories.

> We changed the colour set so that it is easier to distinguish the different scenarios.

**Referee Feedback**:

1. "If this is the case – that prices are not included in the analysis before a certain point (please specify which point), then this has to be mentioned explicitly in the paper."

> Explicitly mentioned in section titles under the results to avoid confusion:
>
> - section 3.1. '*Phosphorus recovery potentials **excluding economic dynamics.**'*
> - section 3.2. *'Phosphorus recovery potentials **including economic dynamics'***

6. "I have not found any explanation or sources of data for labour wage estimates in the Supplementary Materials."

We now explain the origin of the labour wage estimate. This explanation is presented under the transport parameter table (Table S3).

"* 15 [$/h] value is an informed estimate of trucking labour wage. In reality, trucking labour wages will vary globally. The model could not account for spatial diversity in wages provided the absence of a global data set on trucking wages per country or even per region. Instead, the 15 [$/h] was determined taking into consideration the wage indicators from the U.S. (c.a 20 $/h) (United States Department of Labor, 2017), the Netherlands (c.a 30$/h) (MijnZZP, 2018), and China (c.a 5 $/h) (Salary Expert, 2018)."

It seems unnecessary to reference these sources. We leave the decision up to the editor on whether or not to include these since the information they provide are only used by the authors as inspiration and are not directly used by the model or in the text.

12. "My point here was to make a clear link between spatial variation in P, its concentration in wastewater, and population and livestock density - ideally in a single sentence (e.g. in Section 1.1). While this connection may be very obvious to someone working on phosphorus, it would be useful to spell it out for non-experts."

Section 2.1.1 reiterates the importance of spatial variation in P (p. 5 L. 0 to p. 5 L. 3):

[revised manuscript text omitted]

**S3. Parameter Sensitivity Figures**

**5  Crop Parameters**

[Figure]

**Transport Parameters**

[Figure]

**Recovery Parameters**

[Figure]

**Market Parameters** (yearly input data)

[Figure]

**Figure S3. Figures reveal how -50% to +50% change in parameter influences the model determined total quantity of phosphorus traded (1st column), the total quantity of recovered products traded (2nd column), and the optimal market price (3rd column). Green marker shows original parameter value with original output. Yellow marker shows relative change in output (y-axis) vs. relative change in parameter (x-axis).**

**Supplementary Tables**

**S1. Phosphorus Production Estimate Data**

| Population | Throughput (kg P) | | Site | Author |
|---|---|---|---|---|
| Bovine (Dairy) | 25 | | US | Barker et al., 2001 |
| | 17.16 | | NL | Blokland, Luesink, & Jongeneel, 2015 |
| | 17.9 | | NL | CBS, 2014 |
| | 20.8 | | US | Weiss & Wyatt, 2004 |
| | 9.6 | (stabled period) | NL | CBS, 2014 |
| Bovine (Beef) | 11.7 | | NL | CBS, 2014 |
| | 13.3 | | US | Barker et al., 2001 |
| | 5.4 | (stabled period) | NL | CBS, 2014 |
| Bovine (Unsp.) | 10 | | Global | Sheldrick et al., 2003 |
| Poultry (layer) | 1.2 | | US | Barker et al., 2001 |
| | 0.17 | | NL | CBS, 2014 |
| Poultry (broiler) | 0.6 | | US | Barker et al., 2001 |
| | 0.08 | | NL | CBS, 2014 |
| Poultry (Unsp.) | 0.19 | | Global | Sheldrick et al., 2003 |
| Swine (sow) | 6.4 | | NL | CBS, 2014 |
| Swine (Unsp.) | 4 | | Global | Sheldrick et al., 2003 |
| | 4.1 | | US | Barker et al., 2001 |
| | 2.1 | | NL | CBS, 2014 |
| Human | 0.77 | | UK | Gilmour et al., 2008 |
| | 0.2-0.7 | | Global | Mihelcic et al., 2011 |
| | 0.78 | | - | CRC, 2005 |
| | 0.7 | | US | Smil, 2000 |

**Table S1. Annual phosphorus excretion rate by species, per head.**

**S2. Crop Phosphorus Requirement Data**

| Crop | Ky [-] | Water req. [mm/harvest] | Growing Period [days] | $P_2O_5$ Range [kg/ha] | $P_2O_5$ Choice [kg/ha] | P [kg/ha] |
|------|--------|-------------------------|-----------------------|------------------------|-------------------------|-----------|
| Maize | 1.30 | 500-800 | 80⁻180 | 36-50 | 50 | 22 |
| Wheat | 0.55 | 450-650 | 120⁻150 | 27-60 | 40 | 15 |
| Rice[1] | 1.00 | 450-700 | 90⁻150 | 26-50 | 35 | 15 |
| Soybean | 0.90 | 450-700 | 135⁻150 | 35 | 35 | 15 |
| Sorghum | 0.90 | 450-650 | 120⁻130 | 20-40, 40-60 | 40 | 15 |
| Potato | 0.90 | 500-700 | 105⁻145 | 39-80 | 80 | 35 |

**Table S2. Crop Data (FAO, n.d.; IFDC & UNIDO, 1998)**
* * *
[1] Fageria, N.K. The Use of Nutrients in Crop Plants. Google books

**S3. Transportation Cost Data**

| **Sea Transport Component Cost** | | | | **Land Transport Component Cost** | | | |
|---|---|---|---|---|---|---|---|
| **Constant** | | **Value** | **Source** | **Constant** | | **Value** | **Source** |
| $E_W$ | [t d$^{-1}$] | 93.1 | (Počuča, 2006) | $E_L$ | [L km$^{-1}$] | 0.53 | (Nylund and Erkkilä, 2005) |
| $W_W$ | [t] | 2,777 | (Počuča, 2006) | $W_L$ | [t] | 60 | |
| $\bar{V}_W$ | [km d$^{-1}$] | 924 | (Počuča, 2006) | $\bar{V}_L$ | [km h$^{-1}$] | 80 | - |
| $C_F$ | [$ d$^{-1}$] | 9,989 | (Počuča, 2006) | $L_c$ | [$ h$^{-1}$] | 15 | -* |
| | | | | $C_d$ | [$ km$^{-1}$] | 0.5 | |

**Table S3. Constants for transport cost determination equation for (eq. 6)**

*. 15 [$/h] value is an informed estimate of trucking labour wage. In reality, trucking labour wages will vary globally. The model could not account for spatial diversity in wages provided the absence of a global data set on trucking wages per country or even per region. Instead, the 15 [$/h] was determined taking into consideration the wage indicators from the U.S. (c.a 20 $/h) (United States Department of Labor, 2017), the Netherlands (c.a 30$/h) (MijnZZP, 2018), and China (c.a 5 $/h) (Salary Expert, 2018).*

**S4. Yearly variable input data**

| | Global Phosphate Ore extraction cost (World Bank, 2018a)[2] | Global Phosphate Ore production (USGS, 2016)[3] | Food Price Index (FAO, 2018a)[4] | Diesel Fuel Price (U.S. Energy Information Administration, 2018)[5] | Bunker Fuel Price (Institut National de la Statistique et des Etudes Economiques, 2017)[6] |
|---|---|---|---|---|---|
| | [$ t$^{-1}$] | [kt] | [-] | [$ gal$^{-1}$] | [$ t$^{-1}$] |
| 2005 | 42.00 | 147,000 | 118 | 2.402 | 248 |
| 2006 | 44.21 | 142,000 | 127 | 2.705 | 290 |
| 2007 | 70.93 | 156,000 | 161 | 2.885 | 341 |
| 2008 | 345.59 | 161,000 | 201 | 3.803 | 522 |
| 2009 | 121.66 | 166,000 | 160 | 2.467 | 355 |
| 2010 | 123.02 | 181,000 | 188 | 2.992 | 464 |
| 2011 | 184.90 | 198,000 | 230 | 3.840 | 642 |
| 2012 | 185.89 | 217,000 | 213 | 3.968 | 672 |
| 2013 | 148.11 | 225,000 | 210 | 3.922 | 613 |
| 2014 | 110.22 | 218,000 | 202 | 3.825 | 546 |
| 2015 | 117.46 | 241,000 | 164 | 2.707 | 291 |

**Table S4. Model yearly variable input data**
* * *
[2] http://pubdocs.worldbank.org/en/226371486076391711/CMO-Historical-Data-Annual.xlsx
[3] https://minerals.usgs.gov/minerals/pubs/commodity/phosphate_rock/
[4] http://www.fao.org/worldfoodsituation/foodpricesindex/en/
[5] https://www.eia.gov/dnav/pet/pet_pri_gnd_dcus_nus_a.htm
[6] https://www.insee.fr/en/statistiques/serie/001642883

**S5. Fixed Parameters**

| | | | |
|---|---|---|---|
| Yearly population growth rate | 1.22 | [%] | (World Bank, n.d.) |
| Yearly livestock growth rate | 0.8 | [%] | (FAO, 2018b) |
| Yearly agricultural expansion rate | 3 | [%] | [-] |
| Struvite recovery efficiency WWTP | 0.2 | [-] | Derived from (Jaffer et al., 2002) |
| Phosphorus density urine (for intracity transport) | 0.00066 | [-] | (Vinnerås, 2001) |
| Phosphorus density faeces (for intracity transport) | 0.00457 | [-] | (Vinnerås, 2001) |
| Phosphorus density rock phosphate | 0.08 | [-] | 18% $P_2O_5$ rock phosphate |
| Phosphorus density Struvite | 0.14 | [-] | - |
| Phosphorus density compost pellets | 0.01 | [-] | (Cofie and Nikiema, 2012) |
| Phosphorus density DAP | 0.2 | [-] | [-] |
| Price Magnesium Chloride | 250 | [$ $t^{-1}$] | (Seymour, 2009) |
| Scaling maint. savings per mass P recovered | 0.89 | [$ $kg^{-1}$] | (Shu et al., 2006) |
| Intracitiy collection distance | 20 | [km] | [-] |
| People serviced per WWTP | 500,000 | [p] | (Egle et al., 2016) |
| Annual costs Struvite precipitation | 180,000 | [$ $a^{-1}$] | (Egle et al., 2016) |
| Annual costs pelletizing facilities | 20,000 | [$ $a^{-1}$] | [-] |
| Pelletizing cost per mass influent | 30 | [$ $t^{-1}$] | (Masayuki Hara, 2001) |
| Annual costs mines (Inv. cost spread over 10 years) | 3,100,000 | [$ $a^{-1}$] | (World Bank, 2018b) |

**Table S5. Model yearly variable input data**

**S5S6. Livestock Phosphorus in Excrement**

| Global P | Year [-] | Cattle [Mt] | Swine [Mt] | Poultry [Mt] | Livestock Total [Mt] |
|---|---|---|---|---|---|
| This study | 2006 | 11.22 | 3.66 | 2.51 | 17.39 (8.8) |
| Bouwman et al., 2013 | 2000 | - | - | - | 17 |
| Sheldrick et al., 2003 | 1996 | 10.43 | 3.55 | 2.74 | 16.72 |

**Table S5S6. Comparison of estimates of global phosphorus produced in the form of excrement by different livestock types, for different years. Estimates made following slaughter weight methodology proposed by (Sheldrick et al., 2003). In brackets for this study, shows contribution of major sites (production greater than 0.5 [t km$^{-2}$] P and greater than 3 [kt] P total)**

**S7. Table of Sensitivity Analysis**

| Parameter Change | Change in P Market Price with Following Changes in Parameter Value: | | | | | | | | | | Sensitivity |
|---|---|---|---|---|---|---|---|---|---|---|---|
| | -50% | -40% | -30% | -20% | -10% | +10% | +20% | +30% | +40% | +50% | |
| **Crop Parameter** | | | | | | | | | | | |
| P Req. for optimal yield | 6 | 4 | 3 | 2 | 1 | -1 | -1 | -1 | -2 | -2 | 0.39 |
| Optimal yield | -7 | 13 | -2 | -1 | -1 | 1 | 1 | 2 | 3 | 3 | 0.26 |
| Crop sale value | -7 | 13 | -2 | -1 | -1 | 1 | 1 | 2 | 3 | 3 | 0.26 |
| Proportion of fertilizer of production cost | -7 | 13 | -2 | -1 | -1 | 1 | 1 | 2 | 3 | 3 | 0.26 |
| Proportion P-fert. cost of total fert. cost | -7 | 13 | -2 | -1 | -1 | 1 | 1 | 2 | 3 | 3 | 0.25 |
| **Recovery Parameters** | | | | | | | | | | | |
| Recovery Efficiency Struvite Precipitation | 0 | 0 | 0 | 0 | 0 | 0 | 0 | 0 | 0 | 0 | 0.00 |
| P density of raw wastes | 0 | 0 | 0 | 0 | 0 | 0 | 0 | 0 | 0 | 0 | 0.00 |
| P density of recovered products | 0 | 0 | 0 | 0 | 0 | 0 | 0 | 0 | 0 | 0 | 0.00 |
| Magnesium cost for struvite | 0 | 0 | 0 | 0 | 0 | 0 | 0 | 0 | 0 | 0 | 0.00 |
| Scaling maintenance savings | 0 | 0 | 0 | 0 | 0 | 0 | 0 | 0 | 0 | 0 | 0.00 |
| Intracity waste collection distance | 0 | 0 | 0 | 0 | 0 | 0 | 0 | 0 | 0 | 0 | 0.00 |
| People serviced per recovery installation | 0 | 0 | 0 | 0 | 0 | 0 | 0 | 0 | 0 | 0 | 0.00 |
| Annual fixed cost per recovery installation | 0 | 0 | 0 | 0 | 0 | 0 | 0 | 0 | 0 | 0 | 0.00 |
| Annual cost mines operation | 0 | 0 | 0 | 0 | 0 | 0 | 0 | 0 | 0 | 0 | 0.00 |
| **Transport parameters** | | | | | | | | | | | |
| Truck fuel efficiency | 0 | 0 | 0 | 0 | 0 | 0 | 0 | 0 | 0 | 0 | 0.00 |
| Ship fuel efficiency | -12 | -10 | -7 | -5 | -3 | 2 | 5 | 7 | 10 | 12 | 0.24 |
| Truck maximum carry load | 9 | 6 | 4 | 2 | 1 | -1 | -1 | -2 | -3 | -3 | 0.17 |
| Ship maximum carry load | 85 | 33 | 21 | 12 | 5 | -5 | -8 | -11 | -14 | -16 | 1.00 |
| Truck velocity | 2 | 2 | 1 | 1 | 0 | 0 | 0 | -1 | -1 | -1 | 0.05 |
| Ship velocity | 43 | 29 | 18 | 11 | 4 | -4 | -7 | -10 | -12 | -14 | 0.87 |
| Truck labour cost | -1 | -1 | -1 | -1 | 0 | 0 | 0 | 1 | 1 | 1 | 0.02 |
| Truck fixed costs | -3 | -3 | -2 | -1 | -1 | 1 | 1 | 2 | 2 | 3 | 0.06 |
| Ship fixed cost (incl. Labour cost) | -11 | -9 | -7 | -5 | -2 | 2 | 4 | 6 | 9 | 11 | 0.22 |
| **Yearly input data** | | | | | | | | | | | |
| Ore extraction cost | -21 | -17 | -13 | -9 | -4 | 4 | 8 | 13 | 17 | 21 | 0.43 |
| Global ore production | 0 | 0 | 0 | 0 | 0 | 0 | 0 | 0 | 0 | 0 | 0.00 |

| | | | | | | | | | | | |
|---|---|---|---|---|---|---|---|---|---|---|---|
| Food commodity price index | 0 | 0 | 0 | 0 | 0 | 0 | 0 | 0 | 0 | 0 | 0.00 |
| Diesel prices | 0 | 0 | 0 | 0 | 0 | 0 | 0 | 0 | 0 | 0 | 0.00 |
| Bunker prices (ship fuel) | -12 | -10 | -7 | -5 | -3 | 2 | 5 | 7 | 10 | 12 | 0.24 |
| Population growth rate | 0 | 0 | 0 | 0 | 0 | 0 | 0 | 0 | 0 | 0 | 0.00 |
| Livestock expansion rate | 0 | 0 | 0 | 0 | 0 | 0 | 0 | 0 | 0 | 0 | 0.00 |
| Agricultural expansion rate | 0 | 0 | 0 | 0 | 0 | 0 | 0 | 0 | 0 | 0 | 0.00 |

**Table S7. This table presents the results of a sensitivity analysis. It reveals how sensitive the model price predictions are to changes in parameter values (other parameters kept constant). A single 'sensitivity' value is determined for each parameter. A value of '1' indicates that the model is highly sensitive to that value, where a (up to 50%) change in parameter value may result in an equal or greater percent change in model output. A value of '0' indicates that the model price output is insensitive to (up to 50%) changes in parameter value. The table shows that the model is most sensitive to changes in transport parameters. Remarkably, the model market prices are relatively insensitive to changes in recovery costs. This is likely because of the insignificant share of recovered P on the total P market.**